# CEPAE: Conditional Entropy-Penalized Autoencoders for Time Series Counterfactuals

**Tomàs Garriga**                                    *tomas.garriga_dicuzzo@novartis.com*
*Novartis*
*Barcelona Supercomputing Center*

**Gerard Sanz**                                      *gerard.sanz_estape@novartis.com*
*Novartis*

**Eduard Serrahima de Cambra**                       *eduard.serrahima_de_cambra@novartis.com*
*Novartis*

**Axel Brando**                                      *axel.brando@bsc.es*
*Barcelona Supercomputing Center*

**Reviewed on OpenReview:** *https://openreview.net/forum?id=X6lrzqOtQo*

## Abstract

The ability to accurately perform counterfactual inference on time series is crucial for decision-making in fields like finance, healthcare, and marketing, as it allows us to understand the impact of events or treatments on outcomes over time. In this paper, we introduce a new counterfactual inference approach tailored to time series data impacted by market events, which is motivated by an industrial application. Utilizing the abduction-action-prediction procedure and the Structural Causal Model framework, we first adapt methods based on variational autoencoders and adversarial autoencoders, both previously used in counterfactual literature although not in time series settings. Then, we present the Conditional Entropy-Penalized Autoencoder (CEPAE), a novel autoencoder-based approach for counterfactual inference, which employs an entropy penalization loss over the latent space to encourage disentangled data representations. We validate our approach both theoretically and experimentally on synthetic, semi-synthetic, and real-world datasets, showing that CEPAE generally outperforms the other approaches in the evaluated metrics.

## 1 Introduction

Time series counterfactual estimation is an essential tool to understand an event's impact on time series data. It is applied to situations where an event or treatment at a certain point in time alters a time series' trajectory, and consists in inferring, once given the observed post-event data, the counterfactual, i.e., the time series that would have taken place if the event had not occurred. Common applications of time series counterfactuals include finance (Barocas et al., 2020) or healthcare (Prosperi et al., 2020). Although our focus here is on theoretical and empirical aspects, the method was originally developed for an application in the pharmaceutical industry, where counterfactuals are used for business planning. Pharmaceutical companies are notably affected when a drug's patent expires and Loss of Exclusivity (LOE) (Castanheira et al., 2019) takes place, prompting competitors to launch cheaper generic versions of the drug. This usually results in a dramatic decrease in the sales volume of about a 60-70% in the first years (Castanheira et al., 2019), severely affecting the company's revenues. Thus, accurately assessing the market impact of generic drug entries is of great importance. See appendix A for an extended discussion on the industrial application of this work.

Synthetic control methods are widely used for time-series counterfactual estimation (Bouttell et al., 2018; Abadie & Gardeazabal, 2003). For example, Causal Impact (Brodersen et al., 2015) estimates counterfactuals

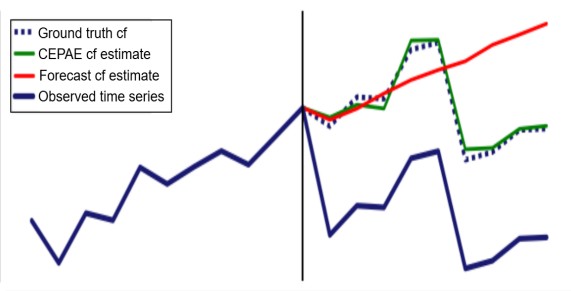

Figure 1: Example of a time series counterfactual (cf) estimation comparison for our synthetic dataset among a baseline LSTM-based forecast model and CEPAE. The vertical line separates pre-event from post-event observations, counterfactual estimations and ground truth.

from two information sources: pre-event observations of the target series and control series that predict the target and are assumed unaffected by the intervention. These methods require suitable control series at inference time. In our industrial setting, such controls are often unavailable or only weakly predictive. A common alternative is to train a forecasting model on historical data and use its predictions as a proxy for the counterfactual, but this ignores information contained in the observed post-event trajectory.

In this paper, we introduce a novel approach for time-series counterfactual estimation that leverages post-event observations of the target series while not requiring any additional time series at inference time, i.e., it extracts post-event information directly from the observed target trajectory. It is rooted in recent advances in causal machine learning (Pawlowski et al., 2020), leveraging the abduction-action-prediction procedure (Pearl, 2000) and encoder-decoder architectures. We instantiate this framework with three encoder–decoder models. First, we adapt a Conditional Variational Autoencoder (CVAE), which has been widely used for counterfactual estimation in image and tabular domains. Second, we consider a Conditional Adversarial Autoencoder (CAAE) inspired by image-translation methods. Finally, we introduce Conditional Entropy-Penalized Autoencoder (CEPAE), which uses an entropy penalty on the latent space to encourage disentanglement from the conditioners. To the best of our knowledge, CEPAE is a new architecture for counterfactual estimation. Figure 1 shows an observed time series that suffers, after the initial drop associated to an event, an additional drop which is not related to it; we see that CEPAE recovers the additional drop in the counterfactual estimate, whereas the LSTM forecast baseline does not. Although our approach requires a sufficient amount of event and event-less time series to train the models, which can be a limitation, it has the advantage of not needing control time series for inference. Having successfully applied our method in our industrial context to infer impacts of generic drug competitors in pharmaceutical industry, in this work we present various experiments that demonstrate its validity.

This paper is structured as follows: Sec. 2 offers a review of the existing literature on counterfactual inference. Sec. 3 provides the necessary background on Structural Causal Models (SCMs) and information theory. Sec. 4 details our methodology, covering the utilized causality concepts, the problem setting and the explanation of our models. In Sec. 5, we introduce the datasets, models and metrics, and discuss the results. Finally, in Sec. 6 we discuss the conclusions.

Our main contributions are: 1) An adaptation of the abduction-action-prediction procedure, which has typically been used with other forms of data such as images, to a time series setting. We have adapted, as well, two autoencoder-based models: CVAE and CAAE. 2) The introduction of CEPAE, an entropy penalized autoencoder for counterfactual estimation, demonstrating its validity both theoretically and experimentally.

## 2 Related Work

Our models are inspired, on the one hand, by SCMs and the abduction-action-prediction procedure (Pearl, 2000), and, on the other hand, on autoencoder (AE) (Bank et al., 2023) and variational autoencoder (Kingma & Welling, 2013) architectures. Parafita & Vitrià (2019) proposes a method for leveraging deep learning

prowess to estimate counterfactuals using the aforementioned Judea Pearl theory. Pawlowski et al. (2020) develops a similar method that specifically uses VAEs. Other related approaches that use autoencoder based architectures are Sanchez-Martin et al. (2021), which use variational graph autoencoders, or Kim et al. (2021), which proposes a VAE-based approach that clusters the causal graph. In appendix B, we provide an extended related work that covers other counterfactual approaches that are not based on autoencoders. On the other hand, most counterfactual estimation works for time series are based on the aforementioned synthetic control and related approaches, which are also covered in appendix B.

It is important to notice that there are several works in the Explainability field that use the term counterfactual in a completely different sense than this work. In the Explainability context, if we consider, for example, a binary classifier and a given input, the term counterfactual refers to the most similar input to the one given that delivers a different classifier outcome. This approach can help understand how the classifier works, but is different from the causal concept of counterfactual. Some works that tackle the Explainability counterfactual problem are applied to time series (Wang et al., 2023; 2021). However, it is important to recognize the difference between this approach and the causal counterfactual problem that our work addresses.

Finally, within the causal inference literature, there is a line of research, represented by works like (Bica et al., 2020; Melnychuk et al., 2022), that addresses a problem called time varying counterfactuals, but the word counterfactual there does not follow the J. Pearl definition, and this line tackles the problem of forecasting outcomes under different temporal sequences of treatments, not the problem of inferring the time series that would have taken place if an event had not occurred.

## 3 Background

In this section, we provide an overview of SCMs and information theory, which are essential for our approach.

### 3.1 Structural Causal Models

The proposed approach for counterfactual estimation is based on the J. Pearl definition of counterfactual (Pearl, 2000), which can be operationalized by employing SCMs.

A SCM $\mathcal{M} := (\mathbf{S}, P(\boldsymbol{U}))$ consists of a collection $\mathbf{S} = \{f_i\}_{i=1}^N$ of structural assignments $a_i := f_i(u_i; \mathbf{pa}_i)$, where $\mathbf{pa}_i$ is the set of parents of $a_i$ (its direct causes), and a joint distribution $P(\boldsymbol{U}) = \prod_{i=1}^N P(U_i)$ over mutually independent exogenous noise variables (i.e. unaccounted sources of variation) (Pawlowski et al., 2020). In a structural causal model (SCM), in general, the assignments are assumed acyclic. Thus, a directed acyclic graph (DAG) can represent relationships, with edges pointing from causes to effects in a causal graph. A unique joint observational distribution $P_{\mathcal{M}}(a)$ is determined by every SCM, fulfilling the causal Markov assumption: each variable is independent of its non-effects given its direct causes. Thus, it factorizes as $P_{\mathcal{M}}(a) = \prod_{i=1}^N (P_{\mathcal{M}}(a_i \mid \mathbf{pa}_i))$, where each conditional distribution $(a_i \mid \mathbf{pa}_i)$ is determined by its assignments and noise distribution (Peters et al., 2017).

SCMs allow to perform counterfactual queries in a three-step procedure (Pearl, 2000): **1) Abduction:** predict the 'state of the world' (the exogenous noise $\boldsymbol{u}$) that is compatible with the observed data $\boldsymbol{a}$, i.e., infer $P_{\mathcal{M}}(\boldsymbol{U} \mid \boldsymbol{a})$; **2) Action:** perform an intervention (e.g. $\text{do}(A_i := \widetilde{a}_i)$) to the counterfactual SCM which corresponds to the desired manipulation, which generates the modified counterfactual SCM $\widetilde{\mathcal{M}} := \mathcal{M}_{\mathbf{a}, do(\widetilde{a}_i)} = (\tilde{\mathbf{S}}, P_{\mathcal{M}}(\boldsymbol{U} \mid \boldsymbol{a}))$; **3) Prediction:** compute the quantity of interest based on the distribution entailed by the modified counterfactual SCM, $P_{\widetilde{\mathcal{M}}}(\boldsymbol{a})$.

### 3.2 Information Theory

For a random variable $X$, we denote as $S(X)$ its entropy, or differential entropy when $X$ is continuous. Considering an additional random variable $Y$, we denote the conditional entropy of $X$ on $Y$ as $S(X|Y)$, and the mutual information among $X$ and $Y$ as $I(X;Y)$. In appendix C, these quantities are properly defined for both discrete and continuous variables, and we discuss some issues regarding differential entropy interpretation. From now on, we will not distinguish among entropy and differential entropy.

For three random variables $X$, $Y$ and $Z$, we present four information theoretic properties, demonstrated in appendix C, that are essential for our development:

$$S(X) = I(X;Y) + S(X|Y), \tag{1}$$

$$S(X,Y) = S(X) + S(Y) - I(X;Y), \tag{2}$$

$$S(X,Y) = S(X) + S(Y|X), \tag{3}$$

$$S(X,Y|Z) = S(X|Z) + S(Y|X,Z). \tag{4}$$

## 4 Method

In this section, we explain our problem setting, the encoder-decoder approach for counterfactual estimation, and the models that we use.

### 4.1 Time Series Counterfactuals: Problem Setting

Next, we define the structure of the time series setting over which our models perform counterfactuals. **Problem Statement.** Let $\boldsymbol{X} = \{\boldsymbol{x}^i\}_{i=1}^N$ be a set of observed time series $\boldsymbol{x}^i = (x_1^i, \ldots, x_T^i)$ with $T$ steps, with $T = T_1 + T_2$. Some of these time series are impacted by an event $e_i$ at time step $T_1 + 1$. Our setting implicitly assumes regularly sampled time series, i.e., a constant time interval between consecutive observations. Time series that are not impacted by the event are also assigned a value $e_i$ that represents the absence of impact. Thus, we can divide each time series into a pre-event segment $h_i = (h_{i1}, \ldots, h_{iT_1})$ with $T_1$ steps and a post-event segment $y_i = (y_{i1}, \ldots, y_{iT_2})$ with $T_2$ steps. From now on, we will denote as $\boldsymbol{H}$, $E$ and $\boldsymbol{Y}$ the variables referring, respectively, to the pre-event time series, the event and the post-event time series, and as $\boldsymbol{h}$, $e$ and $\boldsymbol{y}$ to their specific realizations, often distinguishing among factual values ($e_f$ and $\boldsymbol{y}_f$) and counterfactual values ($e_{cf}$ and $\boldsymbol{y}_{cf}$). For an observation $\left\{\boldsymbol{h}^i, e_f^i, \boldsymbol{y}_f^i\right\}$, our objective is to estimate the counterfactual values $\boldsymbol{y}_{cf}^i$ in the hypothetical scenario where $E$ had taken a different value ($E = e_{cf}^i$) while the rest of the factors of variation of $\boldsymbol{Y}$ were maintained.

Given the previously defined variables, we define a counterfactual function f such that $y_{cf} = \mathrm{f}(\boldsymbol{h}, e_f, \boldsymbol{y}_f, e_{cf})$. Then, our task is to find an approximate counterfactual function $\hat{\mathrm{f}}$ that, similarly to f, estimates $\hat{\boldsymbol{y}}_{cf}$.

In our company, we have access to a large amount of historical sales volume data for many products and countries, a significant amount of which have been impacted by an event. Thus, from these data, we can build a time series dataset of a selected number of steps $T$ which consists of non impacted time series, obtained by applying a $T$ steps rolling window to the non impacted historical data, and impacted time series, obtained by taking, once selected a number of pre-event steps $T_1$ and post-event steps $T_2$ (with $T_1 + T_2 = T$), the windows that match this selection in the impacted historical time series.

In our counterfactual setting, $E$ and $\boldsymbol{H}$ will be the parents of $\boldsymbol{Y}$, and we consider datasets with two structures: $\boldsymbol{H}$ being parent of $E$, becoming a confounder (confounded setting), and no direct relation existing among $\boldsymbol{H}$ and $E$ (unconfounded setting). In our counterfactual setting, $E$ and $H$ are always intervened on ($H$ to its factual value $h$ and $E$ to the counterfactual value $e_{cf}$). Therefore, it is not necessary to treat $E$ and $H$ as random variables generated by exogenous noise terms that must be abducted. This makes counterfactual estimation in both the confounded and the unconfounded settings equivalent at a practical level. Figure 2 shows the computational and the causal graph of our problem. For the models that we present in this work, we assume that our variables data generating process follow a causal graph like the one shown in figure 2, with no hidden confounders, and also positivity.

### 4.2 Encoder-Decoder Based Models for Counterfactual Estimation in our Problem Setting

**Roadmap.** We first describe a generic encoder–decoder instantiation of abduction–action–prediction (Eq. 5 and Box 1), then summarize how CVAE and CAAE implement it, and finally introduce CEPAE and the entropy penalty that enforces the required independence between $Z$ and the conditioners.

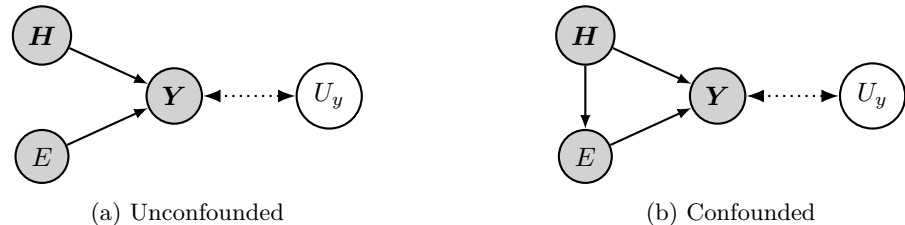

(a) Unconfounded           (b) Confounded

Figure 2: Problem setting's computational graphs. Solid arrows denote direct influences on $Y$; dotted double arrow reflects the abduction link. The confounded panel adds $H{\to}E$.

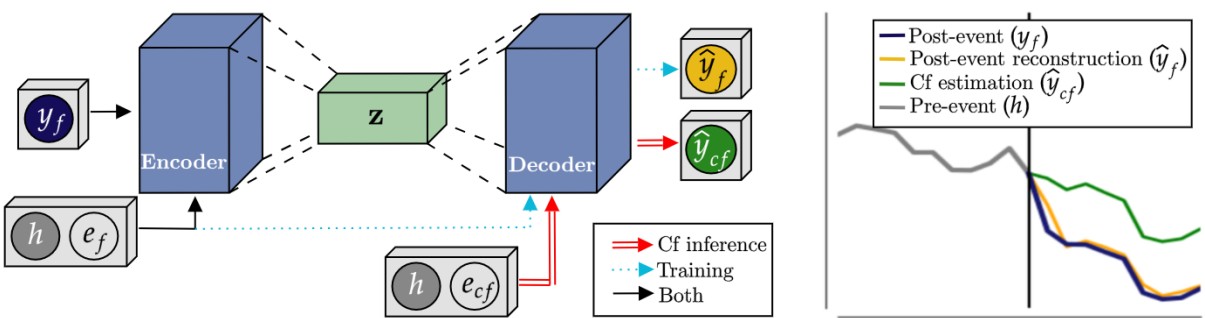

Figure 3: Scheme of encoder-decoder based methods for counterfactual inference, both in training and counterfactual inference phase.

For counterfactual estimation in high-dimensional settings, autoencoder-based architectures are a natural choice as they provide flexible latent representations (Pawlowski et al., 2020; Sanchez-Martin et al., 2021; Lample et al., 2017). In this work, we adapt two such architectures to our time-series setting and propose a new one. Next, we describe how the counterfactuals of our problem setting can be estimated using encoder-decoder architectures and assuming the graphs in figure 2.

Considering the abduction-action-prediction procedure and our counterfactual problem setting, it is possible to define two trainable functions that, together, allow us to estimate a counterfactual $\hat{\boldsymbol{y}}_{cf}$ once given $\boldsymbol{h}$, $\boldsymbol{y}_f$, $e_f$ and $e_{cf}$. We will refer to the first of these functions as an encoder $E_\phi$, with trainable parameters $\phi$, which outputs either the location and scale parameters of a (usually multivariate) latent distribution $P(\boldsymbol{Z}|\boldsymbol{h}, e_f, \boldsymbol{y}_f) = E_\phi(\boldsymbol{h}, e_f, \boldsymbol{y}_f)$, or directly a latent variable $\boldsymbol{z} = E_\phi(\boldsymbol{h}, e_f, \boldsymbol{y}_f)$. $\boldsymbol{Z}$ is a latent representation that should allow the second of our trainable functions, which we call the decoder $D_\theta$ with trainable parameters $\theta$, to estimate counterfactuals $\hat{\boldsymbol{y}}_{cf} = D_\theta(\boldsymbol{z}, \boldsymbol{h}, e_{cf})$. Figure 3 illustrates this process. In terms of the approximated counterfactual function mentioned in 4.1, we can express our functions as:

$$\hat{\boldsymbol{y}}_{cf} = \hat{\mathrm{f}}(\boldsymbol{h}, e_f, \boldsymbol{y}_f, e_{cf}) = D_\theta(E_\phi(h, e_f, \boldsymbol{y}_f), \boldsymbol{h}, e_{cf}). \tag{5}$$

In the case where the encoder outputs the parameters of the latent distribution, the value $\boldsymbol{z}$ will be obtained via sampling from that distribution. For quick reference, we summarize the counterfactual inference procedure in Box 1.

> **Box 1. Method in three steps (abduction–action–prediction).** Given $(h, e^f, y^f)$ and a counterfactual event value $e^{cf}$: **(i) Abduction:** $z \leftarrow E_\phi(h, e^f, y^f)$ (or sample $z \sim p_\phi(z \mid h, e^f, y^f)$); **(ii) Action:** replace $e^f$ by $e^{cf}$ keeping $z$ fixed; **(iii) Prediction:** $\hat{y}^{cf} \leftarrow D_\theta(z, h, e^{cf})$.

There is a direct analogy between this encoder–decoder setting and the abduction–action–prediction procedure. In SCM terms, abduction amounts to inferring a distribution over the exogenous noise $U_y$ given $(\boldsymbol{h}, e_f, \boldsymbol{y}_f)$, i.e., $P_{\mathcal{M}}(U_y \mid \boldsymbol{h}, e_f, \boldsymbol{y}_f)$. Our encoder plays the role of learning a latent variable $\boldsymbol{Z}$ that summarizes this noise: it aims to represent the part of the variation in $\boldsymbol{y}$ that is not explained by $(\boldsymbol{h}, e_f)$.

Table 1: Summary of the three encoder–decoder models used for counterfactual estimation. Here $\boldsymbol{C}$ denotes the conditioners (e.g., $(\boldsymbol{h}, e)$).

| Model | Regularizer on $\boldsymbol{Z}$ | Practical implications |
|---|---|---|
| CVAE | KL divergence with fixed prior $p(\boldsymbol{z})$ | Stable training with standard VAE machinery, but often poor reconstruction and insufficient disentanglement between $\boldsymbol{Z}$ and $\boldsymbol{C}$, which can hurt counterfactual accuracy. |
| CAAE | Adversarial training on $\boldsymbol{Z}$ using an event discriminator. | Encourages disentanglement between $\boldsymbol{Z}$ and $\boldsymbol{C}$ and has good reconstruction, but inherits the instability and tuning difficulties of adversarial training. |
| CEPAE | Entropy penalty $S(\boldsymbol{Z})$. | Simple objective (no discriminator); encourages disentanglement between $\boldsymbol{Z}$ and $\boldsymbol{C}$ and has good reconstruction, having a stable training. |

Importantly, following impossibility results on the identifiability of generative mechanisms with multidimensional exogenous variables (Nasr-Esfahany & Kiciman, 2023), the true exogenous noise $U_y$ is not identifiable in general. Therefore, we do not claim that $\boldsymbol{Z}$ equals $U_y$; instead, we only require that $\boldsymbol{Z}$ is a useful proxy that behaves like such a noise term for the purpose of counterfactual prediction. Then, the decoder (that can be identified with the structural assignment $f_y$) inferring $\hat{\boldsymbol{y}}_{cf}$ would correspond simultaneously to the action and the prediction steps, where we first set the modified SCM $\widetilde{\mathcal{M}} = \mathcal{M}_{\mathbf{h}, \mathbf{e}_f, \mathbf{y}_f; do(E=e_{cf})}$ and then the result $\hat{\boldsymbol{y}}_{cf}$ is obtained as a sample of $\widetilde{\mathcal{M}}$.

**Why disentanglement matters.** If we train a conditional autoencoder only with reconstruction loss, the latent variable $\boldsymbol{Z}$ can encode all the variation in $\boldsymbol{y}$, including the effect of $(\boldsymbol{H}, \boldsymbol{E})$. In that case, the decoder may ignore the conditioners and, at test time, changing $\boldsymbol{E}$ will have little effect on the output. From the SCM in Fig. 2 and the independence of exogenous noises in Sec. 3.1, the variation in $\boldsymbol{y}$ can be decomposed into contributions from its parents $(\boldsymbol{H}, \boldsymbol{E})$ and an exogenous noise term $U_y$ satisfying $U_y \perp \boldsymbol{H}$ and $U_y \perp \boldsymbol{E}$. For counterfactual queries that intervene on $\boldsymbol{E}$, we want the decoder to use $(\boldsymbol{H}, \boldsymbol{E})$ to model the causal effect of the event and to use $\boldsymbol{Z}$ only for the remaining variability. Intuitively, this corresponds to learning a representation $\boldsymbol{Z}$ that is (approximately) independent of $(\boldsymbol{H}, \boldsymbol{E})$ and captures only the variation analogous to $U_y$. In practice, this motivates regularizing the latent space so that $\boldsymbol{Z}$ cannot redundantly encode information already present in the conditioners.

The entropy penalty in CEPAE is designed precisely to encourage this behaviour: Section 4.2.3 provides information-theoretic arguments showing that minimizing the entropy of $\boldsymbol{Z}$ tends to reduce its mutual information with the conditioners while preserving information to reconstruct the input.

Next, we explain three autoencoder based models for counterfactual estimation: CVAE, an autoencoder based model very common, for example, in image counterfactual inference; CAAE, a model inspired by existing works for image manipulation; and CEPAE, a novel counterfactual model which we show that overcomes the previous ones in our time series settings. We only briefly discuss the two first models here, and leave additional information for the appendix. Also, we summarize the core differences between these models in table 1.

### 4.2.1 CVAE

We first apply a conditional VAE, with a KL (Hall, 1987) regularization term, to infer counterfactuals according to the previously defined process. See appendix D for more details. Being CVAE a reasonable option to generate counterfactuals, some relevant problems exist. For example, the model can ignore, completely

or partially, its conditioning $C$, as even if KL divergence limits the capacity of $Z$ to encode information, the model lacks theoretical guarantees of disentanglement of $Z$ with respect to $C$. Some of the techniques that can be used to reduce the capacity of $Z$, such as increasing the weight of KL or reducing the dimensionality of latent space, have severe consequences on the reconstruction capabilities, which also affect the quality of the counterfactual estimations. Even without these techniques, the reconstruction capacity of VAEs is often poor, especially if we compare it to regular AEs (Zhao et al., 2017).

### 4.2.2 CAAE

As mentioned above, regular autoencoders lack mechanisms to disentangle the latent representation from conditioners. On the other hand, their reconstruction capacity far exceeds VAEs. Thus, CAAE is a first attempt to maintain the reconstruction capacity of AEs while furthering disentanglement properties. We take the idea from Lample et al. (2017), a work focused on image manipulation, and adapt it to our counterfactual setting. Based on ideas from domain adaptation (Ganin et al., 2016), the method consists in a conditioned AE where an adversarial training is added that makes the latent representation unpredictive of the value of the conditioner $C$, trying to achieve $Z \perp C$. The adversarial training consists in a model that predicts the value of $C$ from $Z$ while the encoder parameters $\phi$ are trained to achieve the opposite objective. For that, we use gradient reversal layer (Ganin et al., 2016), with an increasing linear scheduling as the weight for the adversarial objective. Alomar et al. (2023) uses a similar idea for counterfactual inference. See appendix E for more details on this model.

Adding adversarial training for several different conditioners can be difficult, especially if they are multidimensional like $H$. In our case, we apply the adversarial training only on $E$, which, in theory, should be enough for our objectives as, for our counterfactuals, we do not intervene on $H$.

In our experiments, CAAE performs well in the unconfounded setting but fails to produce accurate counterfactuals under confounding, despite extensive hyperparameter search. Existing adversarial image-manipulation methods similarly focus on unconfounded settings and do not directly address our causal setup. Moreover, CAAE inherits the usual drawbacks of adversarial training, including instability and higher computational cost (Sridhar et al., 2021); we illustrate these convergence issues empirically and compare them with the behaviour of CEPAE in Appendix M.

### 4.2.3 CEPAE

We propose a conditional autoencoder based model that adds, to the regular reconstruction loss of an AE, an entropy penalty (EP) over the latent representation that reduces its entropy $S(Z)$, minimizing the amount of information that $Z$ can carry.

Intuitively, if the conditioners $C$ are informative about $X$ (i.e., $I(X; C) > 0$), then encoding this information in $Z$ increases the entropy $S(Z)$ without improving the joint information $I(X; Z, C)$ that reconstruction depends on[1]. The entropy penalty therefore discourages $Z$ from redundantly encoding information already available in $C$. Next, we provide a theoretical foundation for this intuition.

Following eq. 1, we can decompose the joint entropy of $Z$ and $C$ in terms of their mutual information with $X$ in the following way:
$$S(Z, C) = I(X; Z, C) + S(Z, C|X). \tag{6}$$
On the other hand, from eq. 2, we can also decompose $S(Z, C)$ in the following way:
$$S(Z, C) = S(Z) + S(C) - I(Z; C). \tag{7}$$
Mixing both and isolating $S(Z)$, we obtain:
$$S(Z) = I(X; Z, C) + S(Z, C|X) - S(C) + I(Z; C). \tag{8}$$
The term $S(Z, C|X)$ in the last equation can be decomposed, following eq. (4), as $S(Z, C|X) = S(C, Z|X) = S(C|X) + S(Z|C, X)$. Finally, we obtain:
$$S(Z) = I(X; Z, C) + S(Z|C, X) + I(Z; C) + S(C|X) - S(C). \tag{9}$$

---

[1] The reconstruction loss aims to increase $I(X; \hat{X})$, and $\hat{X}$ (the reconstruction of $X$) is a function of $Z$ and $C$.

In the last equation, the terms $S(\boldsymbol{C}|X)$ and $S(\boldsymbol{C})$ are fixed from the dataset and do not depend on the model. The term $I(X; \boldsymbol{Z}, \boldsymbol{C})$ would tend to be minimized by the entropy penalty alone, but the reconstruction loss counteracts this by increasing it. Properly weighting the entropy penalty ensures that the influence of the reconstruction loss prevails. The term $S(\boldsymbol{Z}|\boldsymbol{C}, X)$ can be interpreted as a noise term (the amount of entropy of $\boldsymbol{Z}$ which is not predictive of $X$). Finally, $I(\boldsymbol{Z}; \boldsymbol{C})$ is the key term that we aim to minimize by penalizing $S(\boldsymbol{Z})$. Like $S(\boldsymbol{Z}|\boldsymbol{C}, X)$, this term will be minimized with opposition exerted by the reconstruction loss and, since $I(\boldsymbol{Z}; \boldsymbol{C})$ has an inferior limit in 0, the penalization will tend to cancel it, thus aiming for $\boldsymbol{Z} \perp \boldsymbol{C}$. In a deterministic model, $I(X; \boldsymbol{Z}, \boldsymbol{C})$ and $S(\boldsymbol{Z}|\boldsymbol{C}, X)$ can tend, respectively, towards infinity and minus infinity. In appendix F, we further discuss on that and show how this does not affect the meaning and implications of the equation.

***Implementation of the entropy penalty.*** The existing methods for estimating the entropy of continuous variables from a finite number of samples involve complex calculus and usually training specific models. To avoid this, we derive a simple expression that sets an upper bound for the entropy of a continuous multivariate distribution based on the variance of its variables. From equation 2, and taking into account that mutual information can not be negative, we have that $S(\boldsymbol{Z}) = S(Z_1, ..., Z_N) \leq S(Z_1) + .... + S(Z_N)$, where $N$ is the dimensionality of $\boldsymbol{Z}$. On the other hand, there is a theorem that states that, for a single variable, the normal distribution maximizes the differential entropy for a given variance (see the demonstration in appendix C.1 or in Marsh (2013)). Thus, the entropy of a variable $X \sim \mathcal{N}(\mu, \sigma^2)$, which is $S(X) = \log(\sigma\sqrt{2\pi e})$, sets an upper bound to the entropy of an arbitrary distribution with variance $\sigma^2$. This gives us that $S(\boldsymbol{Z}) \leq n \log(\sqrt{2\pi e}) + \log(\sigma_1) + ... + \log(\sigma_N) =: S_{UB}(\boldsymbol{Z})$, where $S_{UB}(\boldsymbol{Z})$ is an upper bound to the entropy of $\boldsymbol{Z}$ based on $\sigma_1, ..., \sigma_N$. Although $S_{UB}(\boldsymbol{Z})$ could, in theory, be used as an entropy penalty loss function, it is unstable due to the logarithm tending to minus infinite for values near zero. We have tried adding small fixed values to the standard deviations inside logarithms but, although some improvements are achieved, model training continues being unstable, slow and suboptimal. Leveraging the fact that any way of reducing standard deviations will reduce $S_{UB}(\boldsymbol{Z})$, we have found that a simple summation over them is more stable and effective as a loss function. Thus, for a batch of data $\{x^k, \boldsymbol{c}^k\}_{k=1}^B$ with $B$ samples, our EP loss is:

$$\mathcal{L}_{\text{EP}}^{\text{batch}}(\phi) = \sum_{i=1}^{N} \sigma_i, \tag{10}$$

where $\sigma_i$ is the standard deviation of the $i$th dimension of the batch of latent representations $\{\mathbf{Z}^k\}_{k=1}^B$, where $\mathbf{Z}^k = \{Z_i^k\}_{i=1}^N$ and $\mathbf{Z}^k = E_\phi(x^k, \boldsymbol{c}^k)$.

On the other hand, the reconstruction loss is:

$$\mathcal{L}_{\text{Rec}}^i(\theta, \phi) = \left\| x^i - D_\theta(E_\phi(x^i, \boldsymbol{c}^i), \boldsymbol{c}^i) \right\|^2. \tag{11}$$

Then, for a batch of size $N_B$, the loss of CEPAE is:

$$\mathcal{L}_{\text{CEPAE}}^{\text{batch}}(\theta, \phi) = \frac{1}{N_B} \sum_{i=1}^{N_B} \mathcal{L}_{\text{Rec}}^i(\theta, \phi) + \lambda \mathcal{L}_{\text{EP}}^{\text{batch}}(\phi). \tag{12}$$

As with CVAE and CAAE, the framework described in beginning of 4.2 allows to generate a counterfactual sample $\hat{\boldsymbol{y}}_{cf}$ by encoding an observation $\hat{\boldsymbol{y}}_f$ and its parents $\boldsymbol{h}$ and $e_f$. Here, we have $\boldsymbol{z} = E_\phi(\boldsymbol{y}_f, \boldsymbol{h}, e_f)$, and then $\hat{\boldsymbol{y}}_{cf} = D_\theta(\boldsymbol{z}, \boldsymbol{h}, e_{cf})$.

Like CAAE, CEPAE has the reconstruction power of regular AEs, which is a clear advantage over CVAE. On the other hand, it has some important advantages over CAAE. First of all, we avoid the instability of adversarial training, using instead an EP to achieve disentangled representations, which is a much simpler regularization technique regarding both computational demands and implementation difficulty. Additionally, although not leveraged in this work, EP implementation offers the advantage over adversarial training of not increasing in complexity with the number of conditioners or their dimensionality.

EP is somewhat reminiscent of the Information Bottleneck theory (Tishby et al., 2000), especially to Strouse & Schwab (2017), although we do not closely follow this framework and the derived disentanglement properties are alien to it. We further discuss this relationship in appendix G. On the other hand, it is important

to mention that works like Nasr-Esfahany & Kiciman (2023) provide an impossibility result for identifiability of generation mechanisms with multidimensional exogenous variables, which affects all counterfactual works applied to multidimensional data. Nonetheless, as Nasr-Esfahany & Kiciman (2023) notes, "exact counterfactual identifiability is often too strong, e.g., in cases where low counterfactual error is tolerable by practitioners". Otherwise, all works on counterfactuals in multidimensional data, like Pawlowski et al. (2020) or Sanchez & Tsaftaris (2022), should be ruled out by the same argument. As we show in the Experimental Section (5), CEPAE's counterfactual errors are low, and deciding if they are tolerable will depend on each application. See appendix H for an extended discussion.

## 5 Experimental Section

In this section, we explain the datasets and the metrics that we have used to validate our models, the details of the models, and finally the results.

### 5.1 Datasets

To evaluate the effectiveness and robustness of our methods, we use several time-series datasets with different characteristics. In causal counterfactual evaluation, real-world datasets rarely provide ground-truth counterfactuals, so direct error metrics are often unavailable. Two common alternatives are: (i) using metrics that assess desirable properties of counterfactuals rather than similarity to ground truth (Sec. 5.2); and (ii) using synthetic or semi-synthetic datasets where the data-generating process is (partly) controlled, enabling ground-truth counterfactuals and standard error metrics. Accordingly, besides our proprietary dataset, we use synthetic and semi-synthetic datasets that mirror our setting of abrupt impact-driven changes while providing counterfactual ground truth. All datasets are split into series with an event at a fixed time step ($e = 1$) and series without an event ($e = 0$). We describe them below.

**Synthetic datasets.** We generate length-30 series starting at 0 with a linear trend sampled uniformly from $[-0.1, 0.1]$ (added at each step). Series with an event include a drop of 0.7 at step 20. In addition, all series (with or without event) undergo an extra change at a random post-event step, sampled uniformly from $[-0.7, 0.7]$, capturing other shocks unrelated to the main event. We add Gaussian noise with variance 0.1 at each step. The only systematic difference between event and non-event series is the drop at step 20, which allows generating paired factual and counterfactual outcomes. We consider two variants: **(1) Unconfounded**, where event assignment is random; and **(2) Confounded**, where event probability follows Bernoulli($p$) with $p = (t + 0.1)/0.2$, where $t$ is the trend.

**Semi-synthetic dataset.** Based on the Rossmann Store Sales dataset (Knauer & Cukierski, 2015) (daily sales, 2013–2015), we simulate an event occurring on the first Monday of March each year that affects half of the stores (e.g., promotion/marketing). The event multiplies sales by 1.1 on day 1, 1.2 on day 2, and 1.3 for the remaining days over three weeks. We use the four weeks before the event as pre-event context and the following three weeks as post-event. This dataset is unconfounded.

**Real-world dataset.** Our private dataset contains monthly sales time series constructed as described in Sec. 4.1. We use 12 months pre-event and 30 months post-event. The dataset is univariate and includes 2310 series with an event and 3000 without. According to business experts, the relationship between $\boldsymbol{H}$ and $E$ is non-existent or very weak.

### 5.2 Evaluated Metrics

To evaluate our methods, we use various metrics. **Mean Absolute Error** (MAE) and **Mean Bias Error** (MBE) are employed to compare estimations with ground truths in synthetic and semi-synthetic datasets, as counterfactual ground truths exist solely for these datasets.

MAE is a direct measure of the accuracy of our estimates and is the most important metric, as it directly compares the counterfactual estimate with the ground truth. On the other hand, MBE, defined as MBE = $\frac{1}{n}\sum_{i=1}^{n}(\hat{y}_i - y_i)$ allows to detect biases. We also consider additional metrics that do not require a ground truth, and hence can be applied to the real world dataset. Each of these metrics assesses some desirable

property of counterfactuals. Overall, these metrics allow us to evaluate counterfactual models even when ground-truth counterfactuals are not available. For completeness, we apply them to all datasets and not only to the real world one. Next, we present the Added Variations metrics, and leave the reconstruction, reversibility and effectiveness metrics from the Axiomatic Definition of Counterfactual (Monteiro et al., 2023) for appendix I due to space restrictions.

**Added Variations.**  We introduce this metric to evaluate how a counterfactual estimation responds to changes in the observations.

The core idea is that, for an accurate method, if we introduce variations in the post-event factual time series, these changes should be treated as effects of processes unrelated to the event and therefore should be reflected in the counterfactual estimate. This metric is implemented as follows: for each time series to be evaluated, several positive and negative values in the order of the data values are chosen; for each of this values, several windows of few consecutive steps from the post-event time series are selected and the chosen value is added to those steps. After that, a counterfactual estimate is obtained for every altered time series and it is compared to the counterfactual estimate of the non-altered time series. Two quantities are obtained: **1) Total difference**, that takes into account the difference among the altered counterfactual and the base counterfactual in all the steps, and **2) Altered steps difference**, which takes into account the difference among the altered counterfactual and the base counterfactual only in the steps affected by the alteration. These quantities are then divided by the expected difference (the product of the alteration value and the number of affected steps). Thus, ideally, the final results for both total difference and altered steps difference metrics should be 1. The final results are obtained by averaging all calculations. For a more formal definition of these metrics, see Appendix J.

**Axiomatic Metrics.**  Following Monteiro et al. (2023) and the Pearlian axioms of counterfactuals (Pearl, 2000; Galles & Pearl, 1998; Halpern, 2000), we summarize three model-agnostic metrics that assess counterfactual soundness without ground-truth counterfactuals. Let $x$ be an observation with factual parents $\mathbf{pa}$, and $x^*$ its counterfactual under parents $\mathbf{pa}^*$. Denote the ideal counterfactual mapping by f, such that $x^* := \mathrm{f}(x, \mathbf{pa}, \mathbf{pa}^*)$, and its learned approximation by $\hat{\mathrm{f}}$.

**Reconstruction (Composition; lower is better).** Null interventions should not change $x$. We measure $\mathrm{Recon} := d_x\Big(x, \hat{\mathrm{f}}(x, \mathbf{pa}, \mathbf{pa})\Big)$, with $d_x$ the Mean Absolute Error (MAE).

**Reversibility (lower is better).** Counterfactual mappings should invert when swapping parents: $\mathrm{Rev} := d_x\Big(x, \hat{\mathrm{f}}(\hat{\mathrm{f}}(x, \mathbf{pa}, \mathbf{pa}^*), \mathbf{pa}^*, \mathbf{pa})\Big)$.

**Effectiveness (higher is better).**  Intervening a parent $pa_k$ to $pa_k^*$ should be reflected in the generated counterfactual. Using a pseudo-oracle $\widehat{\mathrm{Pa}}_k$ that predicts $pa_k$ from an observation, we report $\mathrm{Eff}_k := d_k\Big(\widehat{\mathrm{Pa}}_k\big(\hat{\mathrm{f}}_k(x, pa_k, pa_k^*)\big), pa_k^*\Big)$, with $d_k$ as accuracy for discrete parents (or $\ell_1$ for continuous ones).

In our setting, $x$ is the post-event time series; factual parents are the pre-event series and factual event $(h, e_f)$, while counterfactual parents replace $e_f$ by $e_{cf}$ (keeping $h$). See appendix I for a more complete description.

## 5.3  Models and Baselines

We compare CVAE, CAAE and CEPAE for all the metrics described in 5.2, and we add as a benchmark, for the MAE and MBE comparison with ground truth counterfactuals, an LSTM-based conditional forecast model that has as inputs only $\boldsymbol{H}$ and $E$. Thus, it can be used as a time series counterfactual estimator that does not take into account post-event values. Besides, we add Adversarially Balanced LSTM (AB-LSTM), which can be seen as an adaptation of the models like Counterfactual Recurrent Network (CRN) (Bica et al., 2020) or Causal Transformer (CT) (Melnychuk et al., 2022) to our setting. See appendix K for more details. Metrics like Added Variations only make sense for methods that use the abduction-prediction-action procedure. We omit CAAE metrics from the confounded synthetic dataset because, as mentioned in 4.2.2, it has not been possible to obtain reasonable results.

Although synthetic control methods would be an alternative to our approach, their reliance on the quality of control data does not allow a fair comparison with the SCM based models, which do not use control data. In 5.4 we perform an illustrative synthetic control experiment with the synthetic dataset, where we see that it will or will not outperform CEPAE depending on how predictive the control data are. We see that, in each application, the selection of the optimal model should strongly depend on the control data quality, in case it exists.

The encoder and decoder architectures of CVAE, CAAE and CEPAE are shared, and are based on 1D convolutional and transposed convolutional layers, in a setting inspired by the VAE model for time series generation proposed in Desai et al. (2021). All methods have been implemented with TensorFlow (Abadi et al., 2015), using an Adam optimizer (Kingma, 2014) with a learning rate of $10^{-4}$. Other hyperparameters of CVAE, CAAE and CEPAE such as dimensionality of latent space or the weight of their respective regularizations are particular for each dataset and have been chosen after an optimization process on factual data. For more details about implementation, the code is provided in `https://github.com/tgarriga/CEPAE`. Also, appendix M provides additional convergence plots for CEPAE and CAAE, and appendix N reports a hyperparameter sensitivity analysis for CEPAE.

### 5.4 Results

Table 2 summarizes results across all datasets and both intervention directions (setting 0: $e_f{=}0 \rightarrow e_{cf}{=}1$; setting 1: $e_f{=}1 \rightarrow e_{cf}{=}0$), averaged over 10 seeds. MAE/MBE are only available when counterfactual ground truth exists; hence "–" on the real-world dataset. Figure 4 reports the complementary synthetic-control comparison under varying donor predictiveness for the synthetic dataset.

**Key takeaways.**

- **Accuracy (ground-truth datasets):** CEPAE achieves the lowest cf MAE across all datasets and both intervention directions.

- **Bias:** CEPAE reduces bias relative to CVAE and is best or comparable among abduction–action–prediction models (CVAE/CAAE/CEPAE).

- **Proxy metrics (all datasets):** CEPAE is consistently closest to the ideal Added Variations behaviour and achieves the best or near-best axiomatic scores.

- **Practicality:** removing entropy penalization causes severe degradation (ablation), and CEPAE trains faster than CAAE/CVAE under our setup.

- **Synthetic control:** when strong donors exist, synthetic-control-style baselines can match or outperform CEPAE; as donor quality decreases, CEPAE outperforms them.

**Ground-truth metrics:** Across all synthetic and semi-synthetic datasets and both intervention settings, CEPAE attains the lowest cf MAE, indicating more accurate counterfactual trajectories than the baselines. cf MBE further shows that CEPAE substantially mitigates the systematic bias observed for CVAE; among the abduction–action–prediction models, it is best or comparable across datasets. LSTM and AB–LSTM can occasionally achieve slightly better cf MBE because they are forecasting models (not abduction–action–prediction) and are not biased toward reconstructing the factual post-event series in the same way. For CAAE in the confounded synthetic settings, we report "–" because adversarial training was not stable under our hyperparameter search, and we avoid over-interpreting degenerate counterfactuals.

**Ablation and computational cost.** To isolate the role of entropy penalization, we train the CEPAE architecture without EP regularization on the unconfounded semi-synthetic 1 dataset. This yields cf MAE $0.256 \pm 0.009$ and cf MBE $-0.250 \pm 0.008$, substantially worse than all baselines and strongly biased, supporting that EP is crucial for counterfactual use of the model. We also report wall-clock training times on the unconfounded synthetic dataset (mean over 5 seeds): CEPAE 9:51 $\pm$ 0:33 (min:s), CAAE 11:31 $\pm$ 0:41, CVAE 14:13 $\pm$ 0:23.

Table 2: Combined results across datasets for $e_f$=0 and $e_f$=1 (10 seeds). We evaluate both intervention directions: setting 0 where $e_f$=0 and $e_{cf}$=1, and setting 1 where $e_f$=1 and $e_{cf}$=0. Top block: ground-truth metrics (cf MAE/cf MBE; only available on synthetic and semi-synthetic datasets). Middle: Added Variations (Total/Altered Steps). Bottom: axiomatic metrics (Reconstruction, Reversibility, Effectiveness). "–" indicates not applicable (no counterfactual ground truth) or not reported due to unstable training.

| Metric | Method | Synthetic | | Conf. synth. | | Semi-synth. | | Real world | |
|---|---|---|---|---|---|---|---|---|---|
| | | 0 | 1 | 0 | 1 | 0 | 1 | 0 | 1 |
| **Ground-truth metrics (synthetic and semi-synthetic only)** | | | | | | | | | |
| cf MAE ↓ | LSTM | $.199 \pm .005$ | $.198 \pm .005$ | $.199 \pm .003$ | $.201 \pm .004$ | $.101 \pm .004$ | $.080 \pm .002$ | – | – |
| | AB-LSTM | $.201 \pm .006$ | $.197 \pm .005$ | $.202 \pm .009$ | $.202 \pm .011$ | $.103 \pm .003$ | $.083 \pm .003$ | – | – |
| | CVAE | $.144 \pm .021$ | $.137 \pm .014$ | $.145 \pm .007$ | $.187 \pm .017$ | $.105 \pm .005$ | $.083 \pm .004$ | – | – |
| | CAAE | $.087 \pm .012$ | $.083 \pm .011$ | – | – | $.065 \pm .004$ | $.061 \pm .003$ | – | – |
| | CEPAE | $\mathbf{.066 \pm .007}$ | $\mathbf{.068 \pm .009}$ | $\mathbf{.088 \pm .030}$ | $\mathbf{.086 \pm .016}$ | $\mathbf{.056 \pm .003}$ | $\mathbf{.056 \pm .004}$ | – | – |
| cf MBE $\sim 0$ | LSTM | $\mathbf{.001 \pm .011}$ | $\mathbf{.001 \pm .014}$ | $\mathbf{-.001 \pm .010}$ | $\mathbf{-0.016 \pm .016}$ | $.003 \pm .004$ | $\mathbf{.002 \pm .004}$ | – | – |
| | AB-LSTM | $\mathbf{.001 \pm .015}$ | $\mathbf{.001 \pm .014}$ | $\mathbf{.001 \pm .011}$ | $\mathbf{-.001 \pm .012}$ | $.004 \pm .005$ | $.003 \pm .004$ | – | – |
| | CVAE | $-.067 \pm .019$ | $.067 \pm .024$ | $.051 \pm .017$ | $.126 \pm .028$ | $.011 \pm .010$ | $-.011 \pm .009$ | – | – |
| | CAAE | $-.001 \pm .044$ | $-.018 \pm .021$ | – | – | $-.009 \pm .010$ | $-.008 \pm .008$ | – | – |
| | CEPAE | $.002 \pm .007$ | $.007 \pm .013$ | $.039 \pm .042$ | $-.037 \pm .032$ | $\mathbf{-.001 \pm .004}$ | $\mathbf{.002 \pm .004}$ | – | – |
| **Added Variations (No counterfactual ground truth required)** | | | | | | | | | |
| Total Steps $\sim 1$ | CVAE | $.457 \pm .091$ | $.443 \pm .066$ | $.450 \pm .038$ | $.460 \pm .040$ | $.037 \pm .011$ | $.045 \pm .110$ | $\mathbf{.899 \pm .024}$ | $1.372 \pm .070$ |
| | CAAE | $.910 \pm .060$ | $.897 \pm .066$ | – | – | $.510 \pm .048$ | $.582 \pm .039$ | $.830 \pm .109$ | $1.293 \pm .124$ |
| | CEPAE | $\mathbf{.946 \pm .042}$ | $\mathbf{.931 \pm .061}$ | $\mathbf{.930 \pm .051}$ | $\mathbf{.935 \pm .048}$ | $\mathbf{.747 \pm .052}$ | $\mathbf{.750 \pm .048}$ | $.849 \pm .293$ | $\mathbf{1.183 \pm .098}$ |
| Altered Steps $\sim 1$ | CVAE | $.388 \pm .097$ | $.360 \pm .090$ | $.241 \pm .018$ | $.246 \pm .021$ | $.109 \pm .017$ | $.109 \pm .015$ | $.312 \pm .016$ | $.710 \pm .021$ |
| | CAAE | $.838 \pm .035$ | $.829 \pm .038$ | – | – | $.275 \pm .091$ | $.289 \pm .073$ | $.445 \pm .196$ | $.729 \pm .071$ |
| | CEPAE | $\mathbf{.874 \pm .010}$ | $\mathbf{.833 \pm .055}$ | $\mathbf{.883 \pm .038}$ | $\mathbf{.866 \pm .033}$ | $\mathbf{.300 \pm .015}$ | $\mathbf{.468 \pm .010}$ | $\mathbf{.558 \pm .200}$ | $\mathbf{.794 \pm .111}$ |
| **Axiomatic metrics (no counterfactual ground truth required)** | | | | | | | | | |
| Reconstruction ↓ | CVAE | $.116 \pm .005$ | $.116 \pm .007$ | $.135 \pm .003$ | $.136 \pm .002$ | $.081 \pm .005$ | $.101 \pm .006$ | $.065 \pm .008$ | $.061 \pm .008$ |
| | CAAE | $.055 \pm .004$ | $\mathbf{.056 \pm .002}$ | – | – | $.055 \pm .004$ | $.063 \pm .003$ | $\mathbf{.038 \pm .006}$ | $.044 \pm .007$ |
| | CEPAE | $\mathbf{.051 \pm .006}$ | $.057 \pm .008$ | $\mathbf{.048 \pm .004}$ | $\mathbf{.048 \pm .004}$ | $\mathbf{.045 \pm .004}$ | $\mathbf{.059 \pm .004}$ | $.039 \pm .006$ | $\mathbf{.042 \pm .007}$ |
| Reversibility ↓ | CVAE | $.127 \pm .004$ | $.150 \pm .014$ | $.152 \pm .002$ | $.155 \pm .005$ | $.100 \pm .015$ | $.117 \pm .016$ | $.073 \pm .009$ | $.078 \pm .007$ |
| | CAAE | $.069 \pm .008$ | $.069 \pm .008$ | – | – | $.065 \pm .005$ | $.067 \pm .005$ | $.059 \pm .004$ | $.055 \pm .005$ |
| | CEPAE | $\mathbf{.068 \pm .011}$ | $\mathbf{.063 \pm .009}$ | $\mathbf{.060 \pm .006}$ | $\mathbf{.060 \pm .005}$ | $\mathbf{.050 \pm .004}$ | $\mathbf{.064 \pm .005}$ | $\mathbf{.052 \pm .005}$ | $\mathbf{.054 \pm .004}$ |
| Effectiveness ↑ | CVAE | $\mathbf{1.0 \pm 0.0}$ | $\mathbf{1.0 \pm 0.0}$ | $\mathbf{1.0 \pm 0.0}$ | $\mathbf{1.0 \pm 0.0}$ | $\mathbf{.996 \pm .006}$ | $.991 \pm .004$ | $\mathbf{.631 \pm .005}$ | $\mathbf{.639 \pm .005}$ |
| | CAAE | $.997 \pm .003$ | $.999 \pm .002$ | – | – | $.994 \pm .005$ | $.995 \pm .006$ | $.619 \pm .006$ | $.631 \pm .006$ |
| | CEPAE | $.999 \pm .002$ | $\mathbf{1.0 \pm 0.0}$ | $.999 \pm .001$ | $\mathbf{1.0 \pm 0.0}$ | $.992 \pm .007$ | $\mathbf{.997 \pm .003}$ | $.627 \pm .005$ | $.621 \pm .006$ |

**Proxy metrics: Added Variations and axiomatic metrics.** On Added Variations and the axiomatic metrics, CEPAE is closest to the ideal behaviour in almost all settings and typically achieves the best Reconstruction and Reversibility while remaining comparable in Effectiveness. Together, these metrics indicate that CEPAE's counterfactual mechanism better respects the desired counterfactual properties, including on real-world data where ground-truth counterfactual errors cannot be computed.

**Comparison with Synthetic Control under Controlled Donor Quality** Evaluating CEPAE against external benchmarks is challenging. To the best of our knowledge, this is the first work that applies a deep-learning approach based on SCMs to a time-series counterfactual problem in the Pearlian sense, so there is no directly comparable baseline in this line of work. At the same time, many methods based on synthetic control exist, but comparing to them poses a major challenge. As synthetic control depends on control time series, its performance depends on how predictive they are about the target time series more than in the modeling itself. On the other hand, in order to obtain distance metrics with respect to the ground truth counterfactual, we need synthetic data. If we use synthetic data, we should create ourselves the control data, so the measured performance of synthetic control methods would depend basically on how predictive we decide to make the control time series about the target time series. For this reason, despite being two competing options to solve the same problem, comparing synthetic control with our method is not the best way to evaluate any of them.

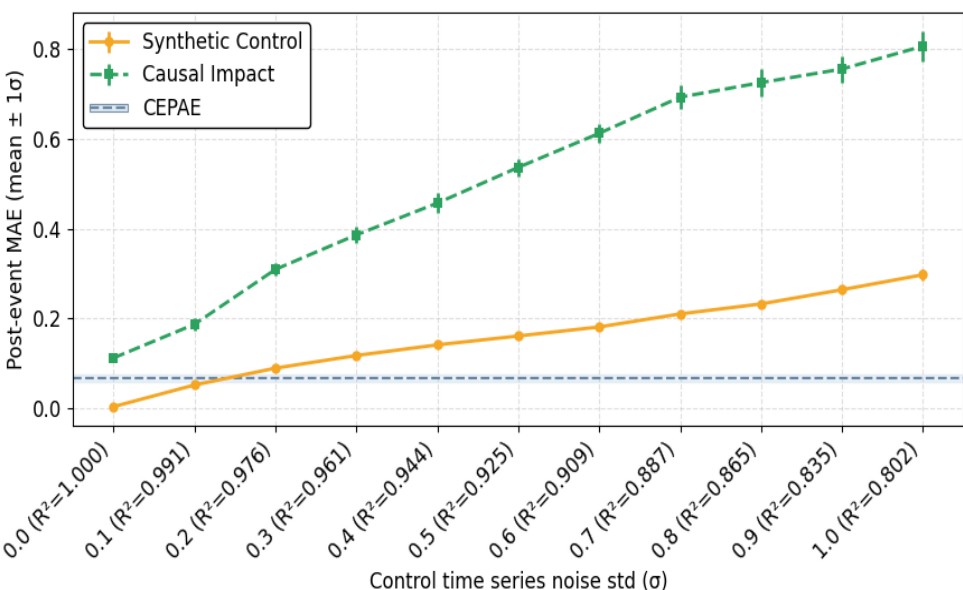

Figure 4: Synthetic Control and Causal Impact comparison.

Despite this, we have performed an experiment of a synthetic control comparison, where we compare CEPAE with several synthetic control scenarios, each having a different level of correlation between control (or donor) and target time series. This experiment is based on the synthetic dataset. We have created, for every "target" time series (i.e., the one over which we want to compute the counterfactual) without observed event ($e_f = 0$), two additional "control" time series or donors (i.e., the time series that we use to predict counterfactual values). Then, we use training data to predict, with an LSTM based model, our target time series from the control time series. This operationalizes the synthetic-control principle—constructing the target from donors with strong pre-period fit—using an LSTM as the combining function, thus being a strong synthetic control baseline. Additionally, we also compare with Causal Impact (Brodersen et al., 2015), a model based on Bayesian structural time series (Almarashi & Khan, 2020). In the case of Causal Impact, a different model is fit for each target time series, whereas in the LSTM-based synthetic-control baseline a single model is trained to approximate all the target series from their corresponding controls. At test time, we evaluate how well the prediction (performed with test data) of the post-event scenario fits the ground truth counterfactual.

We create control time series by adding random Gaussian noise with mean 0 and a specific standard deviation ($\sigma$) to the target time series. With this, we ensure that they are predictive, to some degree, of the target series, being less predictive as the $\sigma$ increases. We report also, for every experiment, the $R^2$ metric of the predicted pre-event target time series from donors.

We show the results in figure 4. Each experiment has been repeated with 10 different random seeds, and corresponds to the 'Synthetic 1' experiment in table 2, where the MAE for CEPAE is $0.068 \pm 0.009$. The first experiment ($\sigma = 0$) corresponds to control time series identical to the target one, therefore being absolutely informative. As expected, the error is very low, almost 0. For $\sigma = 0.1$, a very low degree of noise, the error is slightly lower than CEPAE's error. For $\sigma \geq 0.2$, the error is higher.

This experiment allows to see why a fair general comparison between CEPAE and synthetic control methods can not be established, as results for synthetic control will strongly depend on control data while the SCM based model does not use it. Thus, the selection between our model and synthetic control methods should be problem specific and based on the quality of the control data, in case it is available. In appendix L, we perform a similar experiment with the semi-synthetic dataset. In this experiment, we observe a similar pattern: when the control time series are highly predictive, the synthetic control method outperforms CEPAE; when they are less predictive, CEPAE outperforms synthetic control.

## 6    Conclusion

We have adapted the theory of SCMs and the abduction-action-prediction procedure to time series counterfactual estimation. Three autoencoder based models have been proposed: CVAE, well known in counterfactual literature although not previously applied to time series problems, CAAE, inspired by image manipulation works, and CEPAE, a novel model for counterfactual inference based on an EP. While CVAE shows some superiority with respect to the simple forecast counterfactual estimation in some cases, and CAAE surpasses CVAE in most metrics, CEPAE generally outperforms the other models for most settings. By featuring the reconstruction power of regular AEs instead of VAEs, while not requiring an adversarial training, it makes the abduction-action-prediction process, underexplored in time series counterfactuals, an interesting option for cases similar to ours. The convenience of CEPAE over synthetic control techniques will depend on the availability and informative capacity of control data and the amount of historical event and event-less time series.

## 7    Acknowledgments

We gratefully acknowledge Novartis for sponsoring the industrial PhD. We also thank the Government of Catalonia's Industrial PhDs Plan for funding part of this research. We acknowledge Horizon Europe Programme under the AI4DEBUNK Project (`https://www.ai4debunk.eu`), grant agreement num. 101135757. Additionally, this work has been partially supported by PID2019-107255GB-C21 funded by MCIN/AEI/10.13039/501100011033 and JDC2022-050313-I funded by MCIN/AEI/10.13039/501100011033 by European Union NextGenerationEU/PRTR.

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

## A    Industrial Application

In many industries, knowing the impact of a *competitor entry* on sales is critical for business planning, investment allocation and objectives setting. The pharmaceutical industry is notably affected when a drug's patent expires and Loss of Exclusivity (LOE) (Castanheira et al., 2019) takes place, prompting competitors to launch cheaper generic versions of the drug. This usually results in a dramatic decrease in the sales volume of about 60-70% in the first years (Castanheira et al., 2019), severely affecting company revenues. Thus, accurate assessment of the market impact of generic drug entries is of utmost importance.

*Time series counterfactual estimation* is an essential tool to understand the impact of an event on time-series data. It is applied to situations where an event or a treatment at a certain point in time alters a time series' trajectory, and consists in inferring, once given the observed post-event data, the counterfactual data, i.e., the time series that would have taken place if the event had not occurred[2].

In the case of an event such as the entry in the market of a generic competitor drug, a straightforward approach for counterfactual estimation is to simply use a time series forecasting model trained on historical data which was unaffected by this type of event, and predict the next steps of our pre-event time series according to the model. The main problem of this approach is that its estimations rely solely on pre-event data, lacking the ability to incorporate relevant information of the post-event facts that could have affected the counterfactual time series.

For example, let us imagine that few months after a generic drug enters the market to compete with our target brand, an extreme weather event creates a problem in logistics which results in lower than expected sales in the whole pharmaceutical market. With a forecast-based model we would estimate regular, not reduced, counterfactual sales volume despite the fact that, in all probability, our counterfactual time series would have been affected by the aforementioned happening. This would create a misconception of the impact that would be problematic for business analysis and planning.

In our company, once we have sales observations for a certain number of months, we are interested in knowing what the impact LOE has been. For that, we pass the sequence of post-event observed values to our model, compute the counterfactual, and then obtain the impacts as the difference between the observations and the counterfactuals. This provides a useful information for business planning, and allows to obtain information of impacts that can be used to predict future impacts.

## B    Extended Related Work

Several methods have been proposed for time series counterfactual estimation and causal impact inference in the presence of an event or treatment. Difference in differences (Callaway et al., 2024) is a common approach which requires some control time series that are not affected by the event to estimate counterfactuals. This method features some important limitations like the Parallel Trends Assumption. Synthetic control (Bouttell et al., 2018; Abadie & Gardeazabal, 2003), which generalizes Difference in differences and overcomes some of its limitations, selects several available time series, other than the target one, that have not been affected by the event, computes some weights based on their pre-event similarity to the target time series, and then estimates the counterfactual as the weighted average of the post-event control time series. Causal Impact (Brodersen et al., 2015) is closely related to the synthetic control approach, and its main difference is that it estimates counterfactuals through a model that predicts the target from control series trained with pre-event observations. Matrix Completion Methods (Athey et al., 2021) are an alternative that can be viewed as a combination of synthetic control and the more classical unconfoundedness approach (Rosenbaum & Rubin, 1983; Imbens & Rubin, 2015).

In recent years, many works have appeared that mix the structural causal model (SCM) theory (Pearl, 2000) and deep learning techniques to estimate counterfactuals. Apart from the ones mentioned in the paper, based on VAEs (Kingma & Welling, 2013) and, to a lesser extent, normalizing flows (Kobyzev et al., 2020), other interesting approaches have been proposed. For example, Shen et al. (2021) use GANs (Goodfellow et al.,

---

[2]Note that it is also possible for the opposite scenario: the observation might be when the event doesn't occur, and the counterfactual when it does.

2014), while Jeanneret et al. (2022) uses diffusion models (Ulhaq et al., 2022). Monteiro et al. (2023) presents some useful metrics to evaluate counterfactuals, which are used in our paper to evaluate the models. Sauer & Geiger (2021) uses deep neural networks to disentangle object shape, object texture and background in natural images. Van Looveren & Klaise (2021) utilizes class prototypes in order to find interpretable counterfactual explanations. Parascandolo et al. (2018) uses multiple competing models in order to retrieve a set of independent mechanisms from a set of transformed data points in an unsupervised way.

It is important to notice that there are several works in the Explainability field that use the term counterfactual in a completely different sense than this work. In the Explainability context, if we consider, for example, a binary classifier and a given input, the term counterfactual refers to the most similar input to the one given that delivers a different classifier outcome. This approach can help understand how the classifier works, but is different from the causal concept of counterfactual. Some works that tackle the Explainability counterfactual problem are applied to time series (Wang et al., 2023; 2021). However, it is important to recognize the difference between this approach and the causal counterfactual problem that our work addresses.

There are other interesting works about causal representation learning with deep generative models that do not tackle directly the problem of counterfactual inference but are interesting to take into account. Kocaoglu et al. (2018) and Liu et al. (2019) combine GANs (Goodfellow et al., 2014) with SCMs, basing a generator architecture on an assumed causal graph. However, this method lack tractable abduction capabilities and therefore cannot generate counterfactuals, reaching only the second rung of the causal ladder (Pearl, 2000). Yang et al. (2022) proposes a method that learns a causal model, including the directed acyclic graph (DAG), over latent variables from data, and generates counterfactual samples. Kumar et al. (2023) uses a GAN based approach to address the specific problem of spurious correlations in medical datasets.

Finally, there are other interesting counterfactual estimation approaches that are applied to tabular data. Among them, Yoon et al. (2018), based on GANs, and Vlontzos et al. (2021), based on Deep Twin Networks, similar to Siamese networks (Koch et al., 2015), stand out.

## C  Extended Information Theory Background

The formula for the differential entropy $S(X)$ of a continuous random variable $X$ with probability density function $f(x)$ over its support $\mathcal{X}$ is given by:

$$S(X) = -\int_{\mathcal{X}} f(x) \log f(x)\, dx. \tag{13}$$

The conditional differential entropy $S(X|Y)$ of a continuous random variable $X$ given another continuous random variable $Y$ with joint probability density function $f_{X,Y}(x,y)$ and conditional probability density function $f_{X|Y}(x|y)$ is given by:

$$S(X|Y) = -\int_{\mathcal{X}} \int_{\mathcal{Y}} f_{X,Y}(x,y) \log f_{X|Y}(x|y)\, dx\, dy \tag{14}$$

The joint entropy $S(X,Y)$ of two random variables $X$ and $Y$ with joint probability density function $f_{X,Y}(x,y)$ is given by:

$$S(X,Y) = -\int_{\mathcal{X}} \int_{\mathcal{Y}} f_{X,Y}(x,y) \log f_{X,Y}(x,y)\, dx\, dy \tag{15}$$

The conditional joint entropy $S(X,Y|Z)$ of two random variables $X$ and $Y$ given a third random variable $Z$ with joint probability density function $f_{X,Y,Z}(x,y,z)$ and conditional joint probability density function $f_{X,Y|Z}(x,y|z)$ is given by:

$$S(X,Y|Z) = -\int_{\mathcal{Z}} \int_{\mathcal{X}} \int_{\mathcal{Y}} f_{X,Y,Z}(x,y,z) \log f_{X,Y|Z}(x,y|z)\, dx\, dy\, dz \tag{16}$$

The mutual information $I(X;Y)$ between two continuous random variables $X$ and $Y$ with joint probability density function $f_{X,Y}(x,y)$ and marginal probability density functions $f_X(x)$ and $f_Y(y)$ is given by:

$$I(X;Y) = \int_{\mathcal{X}} \int_{\mathcal{Y}} f_{X,Y}(x,y) \log\left(\frac{f_{X,Y}(x,y)}{f_X(x)f_Y(y)}\right) dx\, dy \tag{17}$$

The mutual information can have values from 0 to $S(X)$ when $X$ and $Y$ are the same distribution.

The conditional mutual information $I(X;Y|Z)$ between two continuous random variables $X$ and $Y$ given a third continuous random variable $Z$ with joint probability density function $f_{X,Y,Z}(x,y,z)$ and conditional probability density functions $f_{X,Y|Z}(x,y|z)$, $f_{X|Z}(x|z)$, and $f_{Y|Z}(y|z)$ is given by:

$$I(X;Y|Z) = \int_{\mathcal{Z}} \int_{\mathcal{X}} \int_{\mathcal{Y}} f_{X,Y,Z}(x,y,z) \log\left(\frac{f_{X,Y|Z}(x,y|z)}{f_{X|Z}(x|z)f_{Y|Z}(y|z)}\right) dx\, dy\, dz \tag{18}$$

All the previous relations are valid also for discrete variables if integrals are changed by summations.

For discrete variables, the entropy is interpreted as a measure of how informative, and uncertain, a random variable is, or as the mean amount of information that its outputs bring (in the case of conditioned entropy, how informative a variable is once conditioned on another variable), and the mutual information is interpreted as the amount of information that one variable brings about another one. Sometimes, the differential entropy is interpreted in the same way as the discrete entropy and, in this work, we do so in an intuitive level. However, there are some limitations with respect to this. For example, the differential entropy can have values lower than 0, and a change of scale can modify it, which is counter intuitive. Nevertheless, the mutual information preserves its meaning for two continuous or one continuous and one discrete variable, which is fundamental for our methodology.

In the main paper, we state these four information theory equations that are essential for the development of CEPAE:

$$S(X) = I(X;Y) + S(X|Y), \tag{19}$$

$$S(X,Y) = S(X) + S(Y) - I(X;Y), \tag{20}$$

$$S(X,Y) = S(X) + S(Y|X), \tag{21}$$

$$S(X,Y|Z) = S(X|Z) + S(Y|X,Z). \tag{22}$$

Relation 21 is the chain rule of entropy, which is an established result in information theory, and relation 22 is the chain rule of entropy in the context of conditional entropy.

Relation 19 is simply a rearrangement of the definition of mutual information:

$$I(X;Y) = S(X) - S(X|Y). \tag{23}$$

To demonstrate relation 20, using the chain rule of entropy, we have:

$$S(X,Y) = S(X) + S(Y|X). \tag{24}$$

From the definition of mutual information:

$$I(X;Y) = S(Y) - S(Y|X). \tag{25}$$

Rearranging this equation, we find:

$$S(Y|X) = S(Y) - I(X;Y). \tag{26}$$

Substituting this back into the chain rule expression for joint entropy, we obtain:

$$S(X,Y) = S(X) + S(Y) - I(X;Y). \tag{27}$$

### C.1 Entropy Maximization by the Gaussian Distribution

**Theorem:** With a normal distribution, differential entropy is maximized for a given variance. A Gaussian random variable has the largest entropy amongst all random variables of equal variance, or, alternatively, the maximum entropy distribution under constraints of mean and variance is the Gaussian (Cover, 1999).

**Proof:** Let $g(x)$ denote a Gaussian probability density function (PDF) with mean $\mu$ and variance $\sigma^2$, and let $f(x)$ be an arbitrary PDF with the same variance. Since differential entropy is invariant under translations, we can assume without loss of generality that $f(x)$ shares the same mean $\mu$ as $g(x)$.

The Kullback–Leibler divergence between these two distributions is given by

$$0 \leq D_{KL}(f\|g) = \int_{-\infty}^{\infty} f(x) \log\left(\frac{f(x)}{g(x)}\right) dx = -S(f) - \int_{-\infty}^{\infty} f(x) \log g(x)\, dx.$$

Next, consider the integral involving $g(x)$:

$$\int_{-\infty}^{\infty} f(x) \log g(x)\, dx = \int_{-\infty}^{\infty} f(x) \log\left(\frac{1}{\sqrt{2\pi\sigma^2}} e^{-\frac{(x-\mu)^2}{2\sigma^2}}\right) dx.$$

This can be expanded and simplified as follows:

$$
\begin{aligned}
\int_{-\infty}^{\infty} f(x) \log g(x)\, dx &= \int_{-\infty}^{\infty} f(x) \log \frac{1}{\sqrt{2\pi\sigma^2}}\, dx + \int_{-\infty}^{\infty} f(x)\left(-\frac{(x-\mu)^2}{2\sigma^2}\right) dx \\
&= -\frac{1}{2}\log(2\pi\sigma^2) - \frac{1}{2}\log(e) \\
&= -\frac{1}{2}\log(2\pi e \sigma^2) \\
&= -S(g).
\end{aligned}
$$

with equality holding if and only if $f(x) = g(x)$, as dictated by the properties of the Kullback–Leibler divergence.

## D  CVAE Description

The loss function for a classic CVAE (Kingma & Welling, 2013) with a $\beta$ penalty (Higgins et al., 2017), which corresponds to the evidence lower bound (ELBO), for a datum $x$ and a conditioning $\mathbf{c}$, is given by:

$$\mathcal{L}_{\text{CVAE}}(\theta, \phi) = \mathbb{E}_{q_\phi(\boldsymbol{z}|x,\mathbf{c})}[\log p_\theta(x|\boldsymbol{z},\mathbf{c})] - \beta\, KL[q_\phi(\boldsymbol{z}|x,\mathbf{c}) \parallel p(\boldsymbol{z}))] \tag{28}$$

where both $q_\phi(\boldsymbol{z}|x,\mathbf{c})$ and $p_\theta(x|\boldsymbol{z},\mathbf{c})$ are a set of dimension-wise independent normal distributions parameterised, respectively, by an encoder neural network $E_\phi$ and a decoder neural network $D_\theta$, $p(\boldsymbol{z})$ is an isotropic normal prior distribution, KL is the Kullback–Leibler divergence (Hall, 1987) and $\beta$ is a penalization over KL (Higgins et al., 2017). In the model training, the ELBO function is maximized with respect to parameters of the neural networks using the re-parametrization trick to sample from the approximate latent posterior: $\boldsymbol{z} = \mu_\phi(x,\mathbf{c}) + \alpha_\phi(x,\mathbf{c}) \odot \epsilon_{\boldsymbol{z}} \sim \mathcal{N}(0,I)$.

Based on the time series setting and the encoder-decoder counterfactual estimation method described in the main paper, it is possible to generate a counterfactual sample $\hat{y}_{cf}$ by encoding an observation $\hat{y}_f$ and its parents $\boldsymbol{h}$ and $e_f$, i.e. obtaining the normal distribution $q_\phi(\boldsymbol{z}|y_f,\boldsymbol{h},e_f)$, where position and scale parameters come from the encoder: $\mu_\phi, \alpha_\phi = E_\phi(y_f,\boldsymbol{h},e_f)$, then sampling the latent posterior from this distribution: $\boldsymbol{z} \sim q_\phi(\boldsymbol{z}|y_f,\boldsymbol{h},e_f)$, and finally decoding it along with the counterfactual event $e_{cf}$: $\hat{y}_{cf} \sim p_\theta(y_f|\boldsymbol{z},\boldsymbol{h},e_{cf})$. Notice that, at a practical level, the counterfactual will be decoded from the latent sample in a deterministic way: $\hat{y}_{cf} = D_\theta(\boldsymbol{z},\boldsymbol{h},e_{cf})$.

## E   CAAE Description

For a data instance $(x^i, \mathbf{c}^i)$, let $G_\psi(E_\phi(x^i, \mathbf{c}^i))$ be a trainable classifier or regressive estimator of $\mathbf{C}$ (depending on whether $\mathbf{C}$ is categorical or continuous) with parameters $\psi$, then we need to use a conditioner loss $\mathcal{L}_{\text{Cond}}^i(\phi, \psi) = \mathcal{L}_{\text{Cond}}^i(\mathbf{c}^i, G_\psi(E_\phi(x^i, \mathbf{c}^i)); \phi, \psi)$, which can take different expressions depending on that kind of conditioner or conditioners we have. In an adversarial game, $G_\psi$ has to maximize this loss and $E_\phi$ has to minimize it. Considering that the reconstruction loss for a conditioned regular AE is:

$$\mathcal{L}_{\text{Rec}}^i(\theta, \phi) = \left\| x^i - D_\theta(E_\phi(x^i, c^i), c^i) \right\|^2 \tag{29}$$

the total loss of CAAE is:

$$\mathcal{L}_{\text{CAAE}}^i(\theta, \phi, \psi) = \mathcal{L}_{\text{Rec}}^i(\theta, \phi) - \lambda \mathcal{L}_{\text{Cond}}^i(\phi, \psi), \tag{30}$$

where $\lambda$ controls the trade-off between the quality of the reconstruction and the invariance of the latent representation. To implement the adversarial training using backpropagation, we use the Gradient Reversal Layer (GRL) (Ganin et al., 2016) and, as in Lample et al. (2017), start off with a $\lambda$ value of 0 and increase it linearly for each iteration.

By using the objective in eq. 30, we should reach a saddle point $(\hat{\theta}, \hat{\phi}, \hat{\psi})$ that achieves the equilibrium between invariance of representation and reconstruction:

$$(\hat{\theta}, \hat{\phi}) = \arg \min_{\theta, \phi} \mathcal{L}_{\text{CAAE}}(\theta, \phi, \hat{\psi})$$
$$\hat{\psi} = \arg \max_{\psi} \mathcal{L}_{\text{CAAE}}(\hat{\theta}, \hat{\phi}, \psi). \tag{31}$$

## F   Further Discussion on Equation 9 terms

In a continuous and deterministic setting (CEPAE is a deterministic model), the conditional distribution $p(\mathbf{Z} \mid \mathbf{C}, X)$ collapses onto a (near) Dirac delta — i.e., $\mathbf{Z}$ is a one-to-one function of $(\mathbf{C}, X)$. As a result, the *conditional differential entropy* $S(\mathbf{Z} \mid \mathbf{C}, X)$ tends to $-\infty$. Simultaneously, the *mutual information* $I(X; \mathbf{Z}, \mathbf{C})$ can diverge to $+\infty$, because knowing $(\mathbf{Z}, \mathbf{C})$ almost determines $X$. Despite these individual divergences, their *sum* in Equation 9 must remain finite *as long as $S(\mathbf{Z})$ itself is finite* — intuitively, the large negative and large positive components cancel each other out in the equation, because the rest of the terms are all finite. Moreover, $S(\mathbf{Z})$ stays finite unless $p(\mathbf{Z})$ degenerates entirely to a delta distribution. This is exactly what would happen if the *entropy penalty* was the only loss, but the *reconstruction loss* in our framework prevents a complete collapse, ensuring that $\mathbf{Z}$ retains some spread and thus retains a finite marginal entropy $S(\mathbf{Z})$. Therefore, while individual terms such as $I(X; \mathbf{Z}, \mathbf{C})$ and $S(\mathbf{Z} \mid \mathbf{C}, X)$ may formally blow up under deterministic mappings, *their sum remains well-defined and finite*, which preserves the overall validity and meaning of Equation 9.

Beyond this, in practice, $\mathbf{Z}$ is stored and processed in finite-precision floating-point format. This means $\mathbf{Z}$ actually takes values from a large but finite set of representable numbers, effectively making it **discrete**. In a *discrete* setting, mutual information *cannot* diverge to $\pm\infty$: there is always an upper bound determined by the (finite) cardinality of the variable's domain, so we do not observe true infinite or negative-infinite entropies. Likewise, conditional entropies in a purely discrete model are nonnegative, ruling out the $-\infty$ scenario. Thus, although we might treat $\mathbf{Z}$ as continuous in theory, real implementations quantize $\mathbf{Z}$ sufficiently to keep the relevant information measures finite.

## G   Information Bottleneck

The Information Bottleneck (IB) framework (Tishby et al., 2000) is an information-theoretical method for learning latent representations. Given an input source $X \in \mathcal{X}$ and a corresponding output target $Y \in \mathcal{Y}$, the goal is to learn an encoder $p_\sigma(Z \mid X)$ that captures only the information in $X$ that is relevant for predicting $Y$. Formally, IB seeks to find sufficient statistics of $X$ with respect to $Y$ using as little information from $X$ as possible. The standard IB objective is expressed as

$$\min_{p_\sigma(Z|X)} \Big( I(X;Z) - \beta\, I(Y;Z) \Big), \tag{32}$$

where $\beta > 0$ is a hyper-parameter balancing how much information from $X$ is retained in $Z$.

Originally, the IB problem was solved with methods like Iterative Blahut–Arimoto algorithms (Yeung & Yeung, 2008), deriving self-consistent equations for the conditional distribution $p_\sigma(Z \mid X)$ (and related marginal distributions) using variational calculus and Lagrange multipliers. This iterative method works well when $X$, $Z$, and $Y$ are discrete and the state spaces are relatively small. However, scaling these methods to high-dimensional or continuous domains is challenging. On the other hand, (Alemi et al., 2017) presented a method to achieve the IB objective with neural networks based on a variational approximation. In fact, VAEs could be seen, with some reservations, as an application of this method, although the original development of VAEs (Kingma & Welling, 2013) does not mention this relationship. A number of works have successfully applied IB for several purposes, like reducing generalization error.

There are important differences between this method and ours. First, IB tries to minimize mutual information among inputs and representations, while our method aims at minimizing the entropy of the representation. Also, our model is deterministic, avoiding the imprecision problems that probabilistic frameworks can show. An idea that is more similar to our method is the one presented in (Strouse & Schwab, 2017). This work proposes a different objective which aims at reducing the entropy of the internal representations instead of the mutual information between the internal representation and the input. The objective can be expressed as:

$$\min_{p_\sigma(Z|X)} \Big( S(Z) - \beta\, I(Y;Z) \Big), \tag{33}$$

If we consider that the objective $Y$ is the same as the input $X$, as in our autoencoder, this objective corresponds, in an information-theoretic sense, to CEPAE's objective. However, there are very substantial differences between this work and ours. First of all, (Strouse & Schwab, 2017) proposes to reach their objective through algorithms similar to those originally used in the IB framework, i.e., they do not employ neural network techniques and therefore the method has important limitations regarding applications to continuous domains. Thus, our implementation of the entropy penalty, and its integration into a neural network model, is distinct from their approach. On the other hand, (Strouse & Schwab, 2017) always consider $Y$ to be different from $X$ and do not mention the possibility of using their framework in an autoencoder architecture.

To the best of our knowledge, no prior approach has explicitly used IB or a similar information-theoretic strategy to disentangle a latent variable from a conditioner to estimate counterfactuals.

## H   Identifiability Limitations in Multidimensional Data

Many works on counterfactual estimation overlook identifiability, mainly because it is usually assumed that, in markovian graphs, all counterfactual queries are identifiable. However, recent works like Nasr-Esfahany & Kiciman (2023) affirm that markovianity alone is insufficient and present more restrictions. In concrete, that paper provides an impossibility result for identifiability of generation mechanisms with multidimensional exogenous variables, which affects all counterfactual works applied to multidimensional data [3], including ours. Nonetheless, as Nasr-Esfahany & Kiciman (2023) notes, "exact counterfactual identifiability is often too strong, e.g., in cases where low counterfactual error is tolerable by practitioners". Otherwise, all works on counterfactuals in multidimensional data, like Pawlowski et al. (2020) or Sanchez & Tsaftaris (2022), should be invalidated. In table 1 of the main paper, we show that our method achieves very reasonable counterfactual errors, and deciding if they are tolerable will depend on each application.

---

[3]Even if a one dimensional latent variable is assumed, that does not lead to identifiability in multidimensional settings (Nasr-Esfahany et al., 2023).

# I  Metrics from the Axiomatic Definition of Counterfactual

As mentioned in our paper, Monteiro et al. (2023) proposes three metrics to measure soundness of a counterfactual inference model without having access to ground truth counterfactuals. It is rooted in the Judea Pearl definition of counterfactual (Pearl, 2000), the soundness theorem (Galles & Pearl, 1998), and the completeness theorem (Halpern, 2000), which, together, state that composition, effectiveness and reversibility are necessary and sufficient properties of counterfactuals in any causal model. Let $x$ be an observation with counterfactual parents $\mathbf{pa}$, and $x^*$ a counterfactual of $x$ with parents $\mathbf{pa}^*$. Then, a counterfactual function f can be defined in such a way that $x^* := \mathrm{f}(x, \mathbf{pa}, \mathbf{pa}^*)$, where the abduction of the exogenous noise $\epsilon$ is implicit. With this notation, where there is a distinction among the ideal counterfactual function f and its approximation with a counterfactual model $\hat{\mathrm{f}}$, Monteiro et al. (2023) defines the axioms that an ideal counterfactual function must obey and propose, in relation to each axiom, a metric to evaluate approximated counterfactual functions. The three metrics are the next ones:

**(1) Composition:**  Intervening on a variable to have the value it would otherwise have without the intervention will not affect other variables in the system. This implies the existence of a null transformation $\mathrm{f}(x, \mathbf{pa}, \mathbf{pa}) = x$ since if $\mathbf{pa}^* = \mathbf{pa}$, then $x$ is not affected. Since the ideal model cannot change an observation under the null transformation, we can measure how much the approximate model deviates from the ideal one by calculating the distance between the original observation and the $m$th time null-transformed observation. Given a distance metric $\mathrm{d_x}$, such as Mean Absolute Error (MAE) (which has been selected in our work), an observation $x$ with parents $\mathbf{pa}$ and a functional power $m$ (which is always 1 in our work), we can measure composition as $\mathbf{Composition}^m := \mathrm{d_x}\left(x, \hat{\mathrm{f}}(x, \mathbf{pa}, \mathbf{pa})\right)$.

**(2) Reversibility:**  Reversibility prevents the existence of multiple solutions due to feedback loops. If a mechanism is invertible, this means that if $x^* := \mathrm{f}(x, \mathbf{pa}, \mathbf{pa}^*)$, then $x = \mathrm{f}(x^*, \mathbf{pa}^*, \mathbf{pa})$. In other words, the mapping between the observation and the counterfactual is deterministic for invertible mechanisms. For a further discussion on this topic, see Monteiro et al. (2023) Thus, it is possible to measure reversibility by calculating the distance between the original observation and the cycled-back transformed observation. Setting $\hat{\mathrm{p}}^{(m)}(x, \mathbf{pa}, \mathbf{pa}^*) := \hat{\mathrm{f}}\left(\hat{\mathrm{f}}(x, \mathbf{pa}, \mathbf{pa}^*), \mathbf{pa}^*, \mathbf{pa}\right)$, given a distance metric $\mathrm{d_x}$, an observation $x$ with parents $\mathbf{pa}$ and a functional power $m$ (which is 1 in our work), we can measure reversibility as $\mathbf{Reversibility}^{(\mathbf{m})}(x, \mathbf{pa}, \mathbf{pa}^*) := \mathrm{d_x}\left(x, \hat{\mathrm{p}}^{(m)}(x, \mathbf{pa}, \mathbf{pa}^*)\right)$. The chosen distance metric in our work is MAE.

**(3) Effectiveness:**  Intervening on a variable to have a specific value will cause the variable to take on that value. Thus, suppose Pa is an oracle function that returns the parents of a variable, then we have the following equality: $\mathrm{Pa}((\mathrm{f}, \mathbf{pa}, \mathbf{pa}^*))$. Effectiveness is difficult to measure objectively without relying on data-driven methods. Following Monteiro et al. (2023), we measure effectiveness individually for each parent by creating a pseudo-oracle function $\hat{\mathrm{Pa}}_K$, which returns the value of the parent $\mathrm{pa}_K$ given the observation. Using an appropriate distance metric $\mathrm{d_k}$, such as accuracy for discrete variables or l1 distance for continuous ones, we measure effectiveness for each parent as $\mathbf{Effectiveness}_k(x, \mathbf{pa}, \mathbf{pa}^*) = \mathrm{d_k}\left(\widehat{\mathrm{Pa}_k}\left(\mathrm{f_k}(x, \hat{\mathrm{pa}}_k, \mathrm{pa}_k^*)\right), \mathrm{pa}_k^*\right)$.

We have given general definitions of these metrics, using generic notation. In our work, the observations correspond to post-event time series, the parents $\mathbf{pa}$ correspond to the factual event and the pre-event time series, and the counterfactual parents $\mathbf{pa}^*$ correspond to the counterfactual event and the pre-event time series.

# J  Added Variations Metrics

The added variations metrics are implemented as follows: for each time series to be evaluated, several positive and negative values in the order of the data values are chosen; for each of this values, several windows of few consecutive steps from the post-event time series are selected and the chosen value is added to those steps. After that, a counterfactual estimate is obtained for every altered time series and it is compared to the counterfactual estimate of the non-altered time series. Two quantities are obtained: **1)**

**Total difference**, that takes into account the difference among the altered counterfactual and the base counterfactual in all the steps, and **2) Altered steps difference**, which takes into account the difference among the altered counterfactual and the base counterfactual only in the steps affected by the alteration. Altered steps difference is the metric reported in the main paper. These quantities are then divided by the expected difference (the product of the alteration value and the number of affected steps). Thus, ideally, the final results for both total difference and altered steps difference metrics should be 1. Next, we give a formal definition of these metrics.

Let $\boldsymbol{y} = \{\boldsymbol{y}_t\}, t \in T$ be the post-event time series over which we want to perform counterfactuals, $\boldsymbol{h}$ its corresponding historical time series previous to the event, $e_f$ the (factual) event, and $\hat{\boldsymbol{y}}_{cf} = \hat{\mathrm{f}}(\boldsymbol{y}, \boldsymbol{h}, e_f, e_{cf})$, where $\hat{\mathrm{f}}$ is a counterfactual function and $e_{cf}$ is the counterfactual event, its corresponding counterfactual estimation. Then, we consider a time series $A = 0...0, v_A...v_A, 0...0$ with $T$ steps, where $v_A$ is the value of the alteration which is added only to a certain number of consecutive steps. Let $\boldsymbol{y}^A = \boldsymbol{y} + A$ be the altered time series, then $\hat{\boldsymbol{y}}_{cf}^A$ would be its corresponding counterfactual estimation. We consider that, if our counterfactual model is correct, alterations in the factual time series should be reflected in the counterfactual time series. Thus, ideally $\sum_i \hat{\boldsymbol{y}}_{cf(i)}^A - \hat{\boldsymbol{y}}_{cf(i)} = \sum_i A_i = n_A \cdot v_A$, where $n_A$ is the number of steps affected by the alteration in $A$. Taking into account that we use different time series $A$ with different values $n_A$ and $v_A$, we can express total differences metric for a single time series $\boldsymbol{y}$ (TD) as:

$$TD = \left\langle \frac{\sum_i \hat{\boldsymbol{y}}_{cf(i)}^A - \hat{\boldsymbol{y}}_{cf(i)}}{n_a \cdot v_A} \right\rangle_A , \tag{34}$$

and altered step differences (ASD) as

$$ASD = \left\langle \frac{\sum_i \hat{\boldsymbol{y}}_{cf(i)}^A - \hat{\boldsymbol{y}}_{cf(i)}}{n_a \cdot v_A} \mathbb{I}_{i \in s_A} \right\rangle_A , \tag{35}$$

where $s_A$ is the set of altered steps (those with value $v_A$ and not 0) in $A$ and $\mathbb{I}$ is the indicator function. We see that, ideally, the result of these averages over the different alteration schemes should be 1. The results given in the paper are the averages of these metrics over all the time series in the test set. The parameters $n_A$, $s_A$ and $v_A$ are particular for every dataset and can be seen in the code.

## K  Adversarially Balanced LSTM

This appendix details the **AB-LSTM** baseline referenced in Sec. 5.3 of the main paper. The model adapts the adversarial–balancing idea of the Counterfactual Recurrent Network (CRN) (Bica et al., 2020) to the single, one–shot event setting studied here.

CRN learns a latent summary that is *predictive* of future outcomes yet *independent* of treatment assignment. We transfer this principle to our scenario, where the treatment/event variable $E \in \{0, 1\}$ occurs once at time $T_0$.

Let the *pre-event history* be $h = (x_1, \ldots, x_{T_0})$, i.e. a realisation of the random variable $H$ defined in Sec. 4.1, and let $e$ denote the binary event flag. AB-LSTM is composed of:

1. *LSTM network* $\phi_\theta$ with hidden $D$; it maps $h \mapsto z \in \mathbb{R}^D$.

2. *Outcome head* $f_\varphi$: a two-layer MLP that predicts the next $T'$ steps, $\hat{y}_{T_0+1:T_0+T'} = f_\varphi(z, e)$.

3. *Event discriminator* $g_\psi$: a logistic classifier that tries to recover $e$ from $z$.

A *gradient-reversal layer* (GRL) (Ganin et al., 2016) with coefficient $\lambda$ is placed between $z$ and $g_\psi$, so gradients flowing to the LSTM network are multiplied by $-\lambda$, forcing $z$ to obscure $e$.

**Training objective.** For a sample $(h, e, y)$ we solve

$$\min_{\theta,\varphi} \max_{\psi} \underbrace{\|f_\varphi(z, e) - y\|_1}_{\text{prediction MAE}} + \lambda \underbrace{\mathrm{BCE}(g_\psi(z), e)}_{\text{adversarial balance}} , \qquad z = \phi_\theta(h).$$

The weight $\lambda$ is ramped linearly from 0 to a $\lambda_{max}$ over the first $40\,\%$ of updates (Ganin et al., 2016).

**Counterfactual generation.** At test time, given factual pair $(h_f, e_f)$,

$$\hat{y}_{\mathrm{f}} = f_\varphi\big(\phi_\theta(h_f), e_f\big), \qquad \hat{y}_{\mathrm{cf}} = f_\varphi\big(\phi_\theta(h_f), 1 - e_f\big).$$

Because no post-event data are required, AB-LSTM acts as a *forecast-only* counterfactual estimator, in the same way as the LSTM baseline. Both of them receive the inputs $(h, e)$, and do not consume post-event values.

## L    Synthetic Control Experiment in Semi-Synthetic Dataset

We perform an experiment similar to the one in the main text with the synthetic dataset. This experiment is based on the semi-synthetic dataset. We have created, for every "target" time series (i.e., the one over which we want to compute the counterfactual) without observed event ($e_f = 0$), two additional "control" time series (i.e., the time series that we use to predict counterfactual values). Then, we use training data to predict, with an LSTM based model, our target time series from the control time series. At test time, we evaluate how well the prediction (performed with test data) of the post event scenario fits the ground truth counterfactual.

We create control time series by adding a random gaussian noise with mean 0 and a specific standard deviation ($\sigma$) to the target time series. With this, we ensure that they are predictive, to some degree, of the target series, being less predictive as the $\sigma$ increases.

Table 3: MAE of the synthetic-control baseline for different noise standard deviations $\sigma$.

| $\sigma$ | MAE |
|---|---|
| 0.0 | $0.004 \pm 0.002$ |
| 0.1 | $0.045 \pm 0.002$ |
| 0.3 | $0.077 \pm 0.001$ |
| 1.0 | $0.114 \pm 0.002$ |
| 1.5 | $0.128 \pm 0.002$ |

We obtain the results in table 3. Each experiment has been repeated with 10 different random seeds, and corresponds to the semi-synthetic 1 experiment in table 1 from the main paper, where the MAE for CEPAE is $0.056 \pm 0.004$. The first experiment ($\sigma = 0$) corresponds to control time series identical to the target one, therefore being absolutely informative. As expected, the error is very low, almost 0. For $\sigma = 0.1$, a very low degree of noise, the error is slightly lower than CEPAE's error. For $\sigma \geq 0.3$, the error is higher.

## M    Convergence Behaviour Comparison between CAAE and CEPAE

In this section, we compare the convergence behaviour of CEPAE and CAAE. The experiments are based on the unconfounded semi-synthetic dataset. We monitor how the counterfactual mean absolute error (MAE), computed using ground-truth counterfactuals on a held-out set, evolves during training. In Figures 5, 6, 7, 8 and 9 we report this counterfactual MAE every 5 epochs, from epoch 0 to 350, under different training regimes.

Figure 5 shows the standard training regime for CEPAE, with the EP weight fixed to 0.09 during the whole training. The model converges slowly but steadily and reaches a fairly stable region around epoch 100. Figure 6 shows the behaviour when we add a linear schedule for the EP weight, analogous to the one used in the main experiments: the weight increases linearly from 0 to 0.09 during the first iterations. In this case, CEPAE converges faster, reaching stable values around epoch 50, with a slight further decrease of the loss afterwards.

Figure 7 shows the behaviour of CAAE when its adversarial weight follows a linear schedule that increases it from 0 to its final value 8.9. The model quickly attains a low MAE around epoch 50, but afterwards the

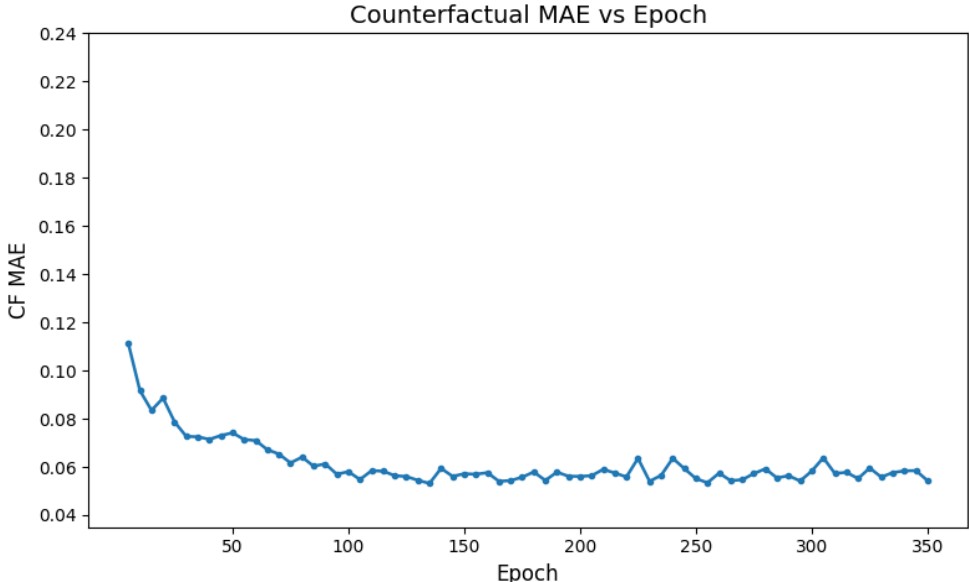

Figure 5: Convergence of CEPAE with a fixed EP weight of 0.09 on the unconfounded semi-synthetic dataset. We plot counterfactual MAE versus training epoch. The model converges slowly but reaches a stable low-error region around epoch 100.

loss grows slowly yet persistently. In figure 8 we show a failure mode that we have occasionally observed for CAAE: at some epoch, the counterfactual MAE suddenly increases and then remains high for the rest of the training. This is an example of adversarial instability.

Finally, Figure 9 illustrates what happens when we do not apply the linear schedule to the adversarial weight in CAAE and instead keep it constant throughout training: the model is unable to reach low-loss regions. In contrast, CEPAE with a fixed EP weight converges more slowly than with a schedule but still reaches good performance after some additional epochs and, importantly, does so without the abrupt instabilities observed for CAAE. This is particularly relevant in real-world applications, where counterfactual ground truths are not available and it is therefore impossible to select the best epoch via early stopping on a validation set based on counterfactual error. The stable convergence of CEPAE makes it more reliable in such settings.

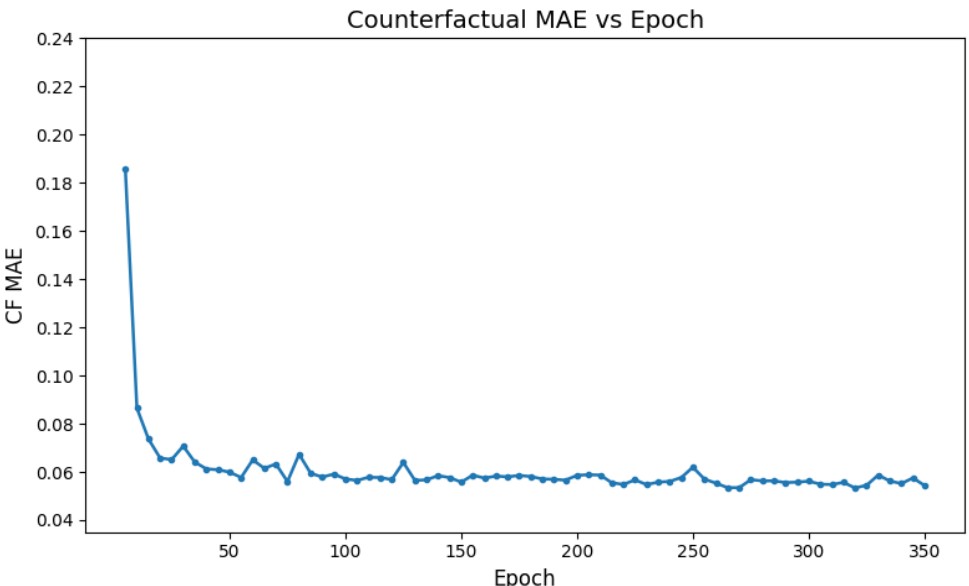

Figure 6: Convergence of CEPAE with a linear schedule for the EP weight on the unconfounded semi-synthetic dataset. The EP weight increases linearly from 0 to 0.09 during the initial epochs. Counterfactual MAE decreases more rapidly than in the fixed-weight case and stabilizes around epoch 50.

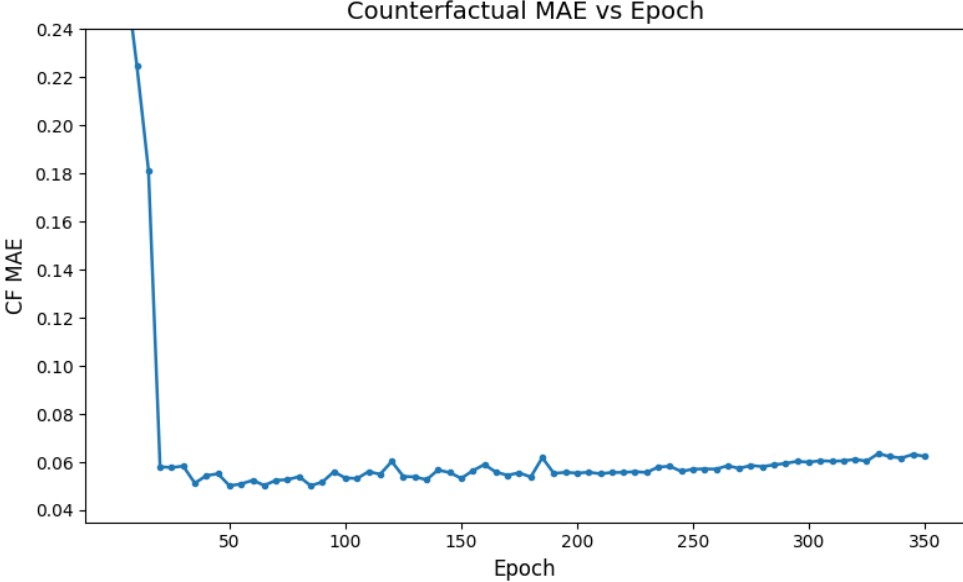

Figure 7: Convergence of CAAE with a linear schedule for the adversarial weight on the unconfounded semi-synthetic dataset. The adversarial weight increases from 0 to 8.9 during training. After an initial drop in counterfactual MAE around epoch 50, the loss slowly increases again, illustrating the instability inherited from adversarial training.

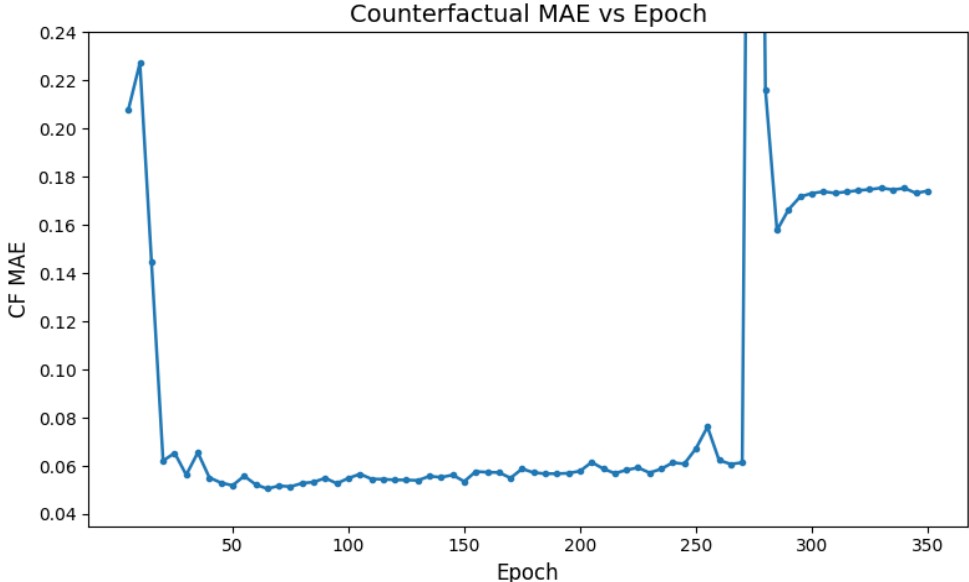

Figure 8: Example of a failure mode for CAAE under a linear adversarial-weight schedule. At some point during training, the counterfactual MAE suddenly increases and remains high for the rest of the epochs, showing a typical instance of adversarial instability.

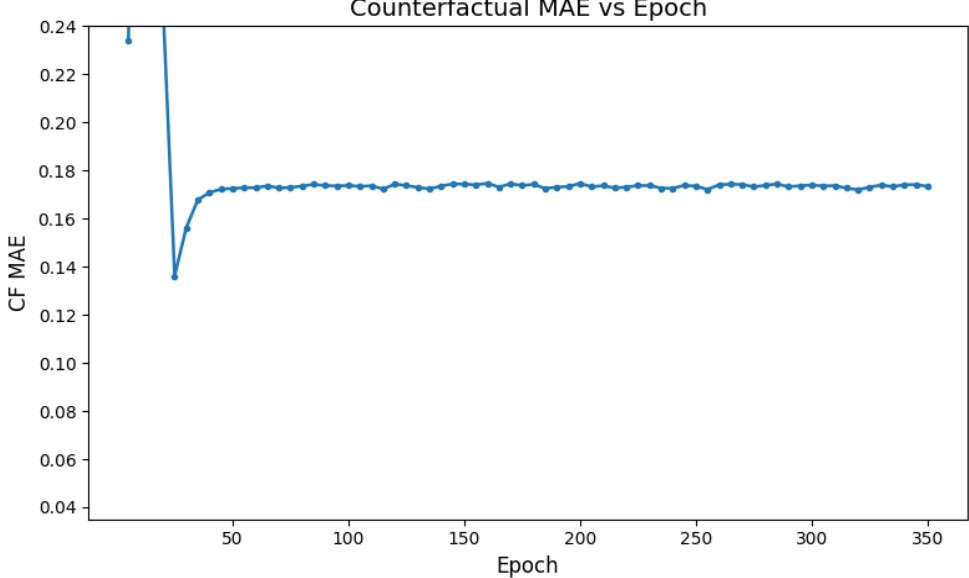

Figure 9: Convergence of CAAE with a fixed adversarial weight of 8.9 on the unconfounded semi-synthetic dataset. Without a schedule, the model fails to reach low-error regions and counterfactual MAE remains high throughout training.

## N   CEPAE Hyperparameter Sensitivity Analysis

In this section, we analyse the sensitivity of CEPAE to its two main hyperparameters: the latent dimensionality and the entropy-penalty (EP) weight. In the experiments reported in the main paper, we use a latent dimensionality of 8 and an EP weight of 0.09. These values were selected after a simple hyperparameter search using the factual data of the unconfounded semi-synthetic dataset.

Here, we study how the counterfactual MAE changes when we vary these hyperparameters. First, we change the latent dimensionality $D_z$ while keeping the EP weight fixed to 0.09. Second, we vary the EP weight $\lambda_{\mathrm{EP}}$ while fixing the latent dimensionality to $D_z = 8$. In both cases we train CEPAE with the same architecture and optimisation settings as in the main experiments and evaluate counterfactual MAE and MBE using the same protocol.

The results are summarised in Tables 4 and 5.

Table 4: Sensitivity of CEPAE to the latent dimensionality $D_z$ on the unconfounded semi-synthetic dataset. We report counterfactual MAE and MBE (mean $\pm$ one standard deviation across seeds). The EP weight is fixed to 0.09.

| Metric | Latent dimensionality $D_z$ | | | | | | |
|---|---|---|---|---|---|---|---|
| | **6** | **7** | **8** | **9** | **10** | **11** | **12** |
| MAE | $0.062 \pm 0.004$ | $0.058 \pm 0.003$ | $0.056 \pm 0.004$ | $0.051 \pm 0.003$ | $0.049 \pm 0.004$ | $0.055 \pm 0.004$ | $0.051 \pm 0.004$ |
| MBE | $-0.008 \pm 0.002$ | $0.002 \pm 0.003$ | $0.002 \pm 0.004$ | $0.006 \pm 0.004$ | $0.010 \pm 0.003$ | $0.006 \pm 0.002$ | $-0.003 \pm 0.002$ |

Table 5: Sensitivity of CEPAE to the EP weight $\lambda_{\mathrm{EP}}$ on the unconfounded semi-synthetic dataset. We report counterfactual MAE and MBE (mean $\pm$ one standard deviation across seeds). The latent dimensionality is fixed to $D_z = 8$.

| Metric | EP weight $\lambda_{\mathrm{EP}}$ | | | | | |
|---|---|---|---|---|---|---|
| | **0.05** | **0.07** | **0.09** | **0.11** | **0.13** | **0.15** |
| MAE | $0.055 \pm 0.004$ | $0.051 \pm 0.003$ | $0.056 \pm 0.004$ | $0.056 \pm 0.003$ | $0.060 \pm 0.004$ | $0.075 \pm 0.004$ |
| MBE | $0.011 \pm 0.004$ | $0.004 \pm 0.003$ | $0.003 \pm 0.004$ | $0.004 \pm 0.004$ | $0.003 \pm 0.002$ | $0.003 \pm 0.002$ |

Overall, we observe that CEPAE is relatively robust to moderate changes in both hyperparameters. For the latent dimensionality, MAE varies smoothly across the range $D_z \in [6, 12]$, with no abrupt degradation in performance and MBE remaining close to zero. Similarly, EP weights in the range $\lambda_{\mathrm{EP}} \in [0.07, 0.11]$ yield very similar MAE and MBE values, while too weak regularisation ($\lambda_{\mathrm{EP}} = 0.05$) or too strong regularisation ($\lambda_{\mathrm{EP}} = 0.15$) leads to a mild deterioration in MAE. These results suggest that the configuration used in the main text ($D_z = 8$, $\lambda_{\mathrm{EP}} = 0.09$) lies in a fairly flat region of the hyperparameter space, and that CEPAE does not require extremely precise hyperparameter tuning to obtain good counterfactual performance.

