# OpenReview forum: "CEPAE: Conditional Entropy-Penalized Autoencoders for Time Series Counterfactuals"
_TMLR — Accepted by TMLR_

### Review · Reviewer_ah7J · 2025-11-05

**Summary Of Contributions:**

This manuscript presents a new framework for counterfactual inference in time series data, introducing the Conditional Entropy-Penalized Autoencoder (CEPAE). The study tests CEPAE on synthetic, semi-synthetic, and real-world datasets, comparing it against established baselines including LSTM forecasts, CVAE, and CAAE.

This paper makes a meaningful contribution to the intersection of causal inference, time series modeling, and representation learning.

**Audience:**

Yes

**Audience Explanation:**

The paper advances the field of time series counterfactual inference with a novel and relevant methodological contribution. I think TMLR's audience would be interested.

**Broader Impact Concerns:**

The paper would benefit from:

broader empirical baselines beyond the autoencoder family,

clearer articulation of interpretability and identifiability implications, and

a more accessible presentation of the underlying causal theory.

**Claims And Evidence:**

Yes

**Claims Explanation:**

The CEPAE framework offers an alternative to variational and adversarial architectures. It demonstrated empirical improvements across several metrics. The theoretical justification for entropy penalization as a route to causal disentanglement is particularly interesting. I think the paper offers sufficient evidence in showing the CEPAE framework.

**Requested Changes:**

While the authors acknowledge the challenges of comparing against synthetic control or causal impact methods, this omission weakens the empirical claim of superiority. Even limited comparisons (e.g., using standard Bayesian Structural Time Series) could provide stronger grounding for CEPAE’s advantages in practice.

The manuscript acknowledges the impossibility of full identifiability in multidimensional counterfactuals (as per Nasr-Esfahany & Kiciman, 2023: https://arxiv.org/abs/2301.09031). Nonetheless, it would benefit from a more explicit discussion on the interpretability of how the learned latent variables relate to domain factors and whether CEPAE’s disentanglement produces semantically meaningful components.

While the paper notes that CEPAE avoids adversarial instability, there is little quantitative discussion of its computational cost, convergence behavior, or hyperparameter sensitivity. These are important factors for reproducibility and industrial scalability.

The real-world evaluation lacks ground-truth counterfactuals, making interpretive confidence somewhat limited. Although the authors employ proxy metrics, a qualitative example or case study illustrating CEPAE’s behavior in an industrial dataset would substantively strengthen the paper’s empirical narrative.

The exposition is dense and assumes substantial background in both causal inference and information theory. Several theoretical sections could be streamlined or visually summarized to improve readability. The core conceptual differences among CVAE, CAAE, and CEPAE could also be more distinctly contrasted.

---

> ### Author Response · Authors · 2025-12-15
>
> We thank the reviewer for taking the time to carefully review our work, for the detailed and constructive suggestions and for the positive overall assessment.
>
> Below we address each of your requested changes.
>
> **1. Broader empirical baselines: synthetic control and Causal Impact**
>
> In the original submission we were already comparing against an **LSTM-based synthetic-control-style baseline**. In the synthetic experiments (Sec. 5, Fig. 4), for each “target” time series (i.e., the series on which we want to compute the counterfactual) we construct additional “control” (donor) time series with controlled levels of pre-event correlation and train an LSTM to predict the target from these donors using only pre-event data. This setup operationalizes the synthetic-control principle—constructing the target from donors with strong pre-event fit—while using an LSTM as the combining function. We vary the strength of the relationship between controls and target to explicitly study performance under different levels of donor predictiveness.
>
> In response to your suggestion, we have now also **added Causal Impact** as an explicit classical baseline in this synthetic-control experiment (see the updated Fig.4). We believe this addition strengthen the empirical grounding of CEPAE’s advantages in practice.
>
> **2. Interpretability of latent factors**
>
> Our primary goal in this work is **not** to obtain axis-aligned, human-interpretable latent components (e.g., a particular dimension of $\boldsymbol{Z}$ representing the trend, another one the level of noise, etc.), but rather to learn a latent representation that behaves like the exogenous noise U_y  in the SCM and is **useful for counterfactual prediction.** When we talk about “disentanglement”, we refer to independence between $\boldsymbol{Z}$ and the event (and more generally the conditioners $(\boldsymbol{H},E)$), not between the different components of $\boldsymbol{Z}$. For this reason, we do not expect to find interesting and interpretable patterns in $\boldsymbol{Z}$. While this is an interesting line for future work, it is beyond the scope of this paper.
>
> **3. Computational cost, convergence behaviour, and hyperparameter sensitivity**
>
> We thank the reviewer for these suggestions, and we have expanded the empirical analysis along all three axes:
>
> 1. **Convergence behaviour**
>
>     We added a dedicated appendix (Appendix M) comparing the convergence behaviour of CEPAE and CAAE on one of our datasets. We monitor **counterfactual MAE** over epochs (on a held-out set with ground-truth counterfactuals) under different training regimes:
>
>     - CEPAE with a fixed EP weight vs a linearly increasing EP weight.
>     - CAAE with a linearly increasing adversarial weight vs a fixed adversarial weight.
>
>         We observe that CEPAE converges steadily to a stable low-error region in both regimes, while CAAE can exhibit typical adversarial-instability patterns: slow but persistent degradation after an initial minimum, and occasional abrupt jumps to much worse MAE from which the model does not recover. We highlight that this is particularly problematic in real-world settings without access to counterfactual ground truths, where early stopping on counterfactual error is not possible, and that CEPAE’s stable convergence is an advantage in such scenarios.
>
> 2. **Hyperparameter sensitivity (Appendix N).**
>
>     We added a **hyperparameter sensitivity analysis** for CEPAE, focusing on its two main hyperparameters: the latent dimensionality $D_z$ and the EP weight $\lambda_{\mathrm{EP}$. We systematically vary:
>
>     - $D_z$ $\in {6,7,8,9,10,11,12\}$ with $\lambda_{\mathrm{EP}}$= 0.09, and
>     - $\lambda_{\mathrm{EP}} \in \{0.05,0.07,0.09,0.11,0.13,0.15\}$ with $D_z$ = 8,
>
>         and report counterfactual MAE and MBE (mean ± standard deviation across seeds). The results show that CEPAE’s performance is **quite stable** across these ranges: MAE varies smoothly, MBE remains close to zero, and the configuration used in the main experiments (D_z = 8, $\lambda_{\mathrm{EP}} = 0.09)$ lies in a region where performance is close to the best observed values. This supports the claim that CEPAE does not require finely tuned hyperparameters to achieve good counterfactual performance.
>
> 3. **Computational cost.**
>
>     In Sec. 5 we now report **average wall-clock training times** over five seeds on the semi-synthetic dataset for the three autoencoder-based models. Under our common architecture and training setup, CEPAE requires approximately $9{:}51 \pm 0{:}33$ (min:s), compared to $11{:}31 \pm 0{:}41$ for CAAE and $14{:}13 \pm 0{:}23$ for CVAE. Thus, CEPAE not only avoids adversarial instability but also trains slightly faster than both CAAE and CVAE in our experiments.
>
>
> We hope these additions address your concerns about convergence, robustness, and computational cost, and strengthen the case for CEPAE’s reproducibility and suitability for industrial deployment.

---

> > ### Author Response · Authors · 2025-12-15
> >
> > **Real-world evaluation and qualitative industrial behaviour**
> >
> > We fully agree that the lack of ground-truth counterfactuals in the real-world dataset limits how far one can go in terms of quantitative claims. In the main text we therefore already focus on **proxy metrics** designed to evaluate desirable properties of counterfactuals without access to ground truths.
> >
> > In the revised version, we have expanded the **industrial application appendix** to describe in more detail how CEPAE is used in practice. Specifically, once a sufficient post-LOE period is observed for a given drug, we feed the post-event trajectory to CEPAE, compute the counterfactual trajectory, and then quantify the LOE impact as the difference between the observed and counterfactual series over time. These impact estimates are used for business planning (e.g., quantifying the revenue loss attributable to LOE) and can be aggregated across products and markets to help forecast expected impacts of future LOE events on similar drugs.
> >
> > Given the proprietary nature of the underlying industrial data, we chose not to include detailed plots of individual product trajectories, but we now make the **practical usage** of CEPAE in this context explicit and connect the model outputs directly to decision-making in the application domain. We believe this additional description helps bridge the gap between the proxy metrics used in the main text and the qualitative way the model’s behaviour is interpreted by domain experts.
> >
> > **5. Exposition, causal theory, and model contrasts**
> >
> > We have taken steps to make the exposition more accessible:
> >
> > - We have **tightened the prose** throughout the main sections to reduce overly long sentences and colloquial phrasing, as also requested by another reviewer.
> > - We introduced a **summary table (Table 1)** that concisely compares CVAE, CAAE, and CEPAE in terms of how they regularize the latent representation \boldsymbol{Z}, and practical implications of respective regularizations (stability, reconstruction, disentanglement). This table directly addresses your request for a clearer contrast between the three models.
> >
> > Combined, these changes aim to make the causal and information-theoretic ideas more approachable while preserving the necessary technical content.
> >
> > Once again, we thank you for your thoughtful review and for highlighting these important aspects. We believe that the revisions significantly strengthen the manuscript.

---

> ### Comment · Reviewer_ah7J · 2025-12-19
> **Good revision!**
>
> Thanks authors for the thorough response. I appreciate the care with which you addressed each point and the clarity of the revisions.
>
> I think the addition of causal impact as a baseline will strengthen the empirical comparison. Also, expanding the discussion with convergence behavior and hyperparameter sensitivity will be more convincing.
>
> Clarification on the role of the latent representation is also helpful and resolves my earlier concern about interpretability.
>
> I also value the expanded description of the industrial use case. I think linking model outputs to decision-making can improve the practical relevance.
>
> Overall, the revisions materially strengthen the manuscript. Great work and I lean towards acceptance!

---

> > ### Author Response · Authors · 2025-12-20
> >
> > Many thanks for taking the time to review the revision and for the encouraging feedback. We’re pleased the changes resolved the points you raised, and we appreciate your support.

---

### Review · Reviewer_WBjb · 2025-11-19

**Summary Of Contributions:**

- The authors formalize the problem of estimating counterfactuals in time series.
- They propose a new type of autoencoder with a penalty that minimizes an upper bound on the entropy of its latent space which should encourage a disentanglement beneficial for reconstructing the counterfactuals.
- They evaluate their method, along with several existing methods, on a set of experiments.

**Audience:**

Yes

**Audience Explanation:**

I believe the problem setting is of interest to researcher in the field, and especially counterfactual estimation (in the Pearlian sense) have been less studied.

**Broader Impact Concerns:**

No remarks

**Claims And Evidence:**

No

**Claims Explanation:**

Looking at the two contributions from the authors, I've some doubts about whether they provide sufficient evidence to support their claims around these two contributions; below I've tried to summarise my main concerns.

## Contribution 1: adapting the abduction-action-prediction procedure to a time series setting.
While I'm not familiar enough with the literature to assert that the authors are not the first to do this, I'm not fully convinced by how they handle the time aspect of the problem, or how their setup differs meaningfully from a non–time-series setting. If I inspect the graphs in Figure 2, there is no time-series structure in it. I would expect something more dynamic, such as nodes in H having arrows pointing to themselves, which could lead to temporal dependencies not currently depicted.  It also seems like a strong assumption that the event always occurs at the same time point or that the time series has the same length always. This issue is also reflected in the model itself, since I don't understand whether the autoencoders take the temporal nature into account (for instance, it would have felt more natural to use some sort of autoregressive model instead). In the end this concern comes down to one thing: could I not simply think of $H$ and $Y$ as multivariate random variables and ignore the time structure, or is there more to it?

## Contribution 2: Demonstrating the validity of CEPAE both theoretically and experimentally
- **For the theoretical part**: What are now actually the theoretical claims in the paper? One key claim by the authors is that we need an disentangled latent representation, but it would be helpful then to see some more arguments (or references to prior work in e.g. the non-time series setting) for why this would solve the problem. And you do not provide theoretical guarantees, I would have hoped for more empirical evidence to show that this disentanglement is necessary. A smaller point, but one that feels glossed over as well. You mention that disentanglement is enabled by minimising the entropy, but ultimately the method minimises an upper bound on the entropy. Under what conditions is minimising this upper bound effective? Are there scenarios where the procedure might fail?
- **For the experimental part**: I'm mainly missing the ablation study which relates to the claim that the disentanglement is critical for doing the counterfactual estimation well, but I've also some other major remarks about your analysis of the experiments which I'm sharing under "Requested changes".

**Requested Changes:**

## Critical adjustment (to secure recommendation for acceptance)
- I would like the authors to establish whether the observed improvements of CEPAE over, e.g., CVAE come from the entropy penalty rather than from the reconstruction loss (which is the main argument for not using CVAE). I believe this would require another experiment where you remove the entropy penalty and compare against your current implementation.
- It is not always clear to me what question you are trying to answer with the metrics used in your experiments, and what reasoning led you to your conclusions. I would hence want a bit more elaborate discussion around the results in Table 1 (more than a single paragraph), because the results seem more nuanced than that CEPAE simply outperforms the other methods. For example, the table shows many interesting trends, yet you state in the results section that “MAE is the most important one,” which raises the question: what is the purpose of the other metrics? For instance, your method sometimes performs worse in terms of MBE (which, by the way, I cannot find a definition for in the paper). There are also some notable differences between the synthetic dataset and the real-world dataset for the axiomatic metrics; I encourage you to comment on why these differences arise. And why are some values for CAAE missing in Table 1? And since you are anyways considering so many metrics, why do you omit mean squared error which seems like a natural choice?
- I would appreciate more elaboration on the setting, particularly what makes your method unique to time series. Could I not simply think of $H$ and $Y$ as multivariate random variables and ignore the time structure, or is there more to it?

## Adjustment which I think could improve the work
- Regarding your claims about minimising the upper bound of the entropy, I would appreciate a discussion of the conditions under which this approach is effective, and when it might fail.
- I understand that the real-world dataset is proprietary, but it would be helpful to provide at least some information about it (such as sample size, dimensionality, etc.) to give readers a better sense of how it relates to the other datasets. Ultimately, what question are you trying to answer by applying your method to this dataset if you do not have access to ground truth?
- I understand that the real-world dataset is proprietary, but it would be helpful to have a bit more information about it such as sample size, dimensionality and so on to give a bit more of an idea how it relates to the other datasets. Ultimately what is the question you're trying to answer by applying your method to this dataset if you don't have a ground-truth?
- I would appreciate some more explanation of the causal graphs in Figure 2: what do the edges between H and Y represent since these are time series? Do all nodes inside $H$ have arrows going into all nodes in $Y$? Can nodes in $H$ also point to other nodes in $H$?
- It would be helpful if you can further elaborate why only autoencoders are considered in this work? You state these are the “most extended and effective,” but it is unclear what this means.

---

> ### Author Response · Authors · 2025-12-15
>
> We thank the reviewer for the careful reading of our manuscript and for the detailed, thoughtful comments. We appreciate in particular the clear structuring of the concerns around (i) the time-series formulation and causal graphs, (ii) the role and necessity of disentanglement and the entropy penalty, and (iii) the interpretation of our experimental metrics and results. In what follows, we address each of these points.
>
> **Time-series handling and causal graphs**
>
> We agree with the reviewer that, at the level of the causal graph in Figure 2, we do not introduce an explicit time-series structure with separate nodes $H_t$, $Y_t$ and temporal self-loops. Instead, $H$ and $Y$ are modeled as block variables corresponding to the full pre-event and post-event trajectories. This is a deliberate modeling choice rather than an oversight.
>
> Our setting is the standard single-event, event-aligned time-series counterfactual problem: there is a unique event at a known time, and our causal query is about the entire post-event trajectory $Y$ conditional on the entire pre-event trajectory $H$ and on the event $E$. For this query, the relevant causal structure is that $(H,E)$ jointly influence $Y$, with $H$ fully observed and never intervened upon. Adding intra-series temporal edges to the graph would not change the interventional distribution we target; it would only make explicit the internal parameterization of the structural equation for $Y$.
>
> Concretely, the temporal dependencies are handled inside the structural equation $Y=f_{Y}(H,E,UY)$, which we parameterize with a convolutional network along the time axis. Thus, the SCM is defined at the segment level because our interventions and queries are segment-level, while the temporal order within each block is captured by the architecture implementing $f_y$.
>
> Formally, $H$ and $Y$ are indeed multivariate variables, but they are time-ordered trajectories, and $f_Y$ is parameterized with a 1D convolution along the time axis, so the model explicitly exploits the temporal ordering even though the causal graph is drawn at block level.
>
> We believe the work remains valuable with this block-level formulation. Beyond the entropy-penalized autoencoder novelty, one of our contributions is to show that the abduction–action–prediction procedure can be leveraged, even without including specific temporal components in the causal graph, to address the single-event time-series counterfactual problem, providing a clear bridge between Pearlian counterfactual semantics and time-series counterfactuals literature. To the best of our knowledge, this specific combination – treating pre- and post-event trajectories as block variables in an SCM, applying abduction–action–prediction to this one-event setting– has not been explored before. We see this as a useful step towards bringing formal causal reasoning into widely used “impact-of-an-event” time-series applications.
>
> **Event timing & real-world dataset**
>
> We agree with the reviewer that it would be beneficial to include more information about the real world dataset. In 4.1, we say:
> ”In our company, we count on a large amount of historical sales volume data for many products and countries, a significant amount of which have been impacted by an event. Thus, from these data, we can build a time series dataset of a selected number of steps $T$ which consists of non impacted time series, obtained by applying a $T$ steps rolling window to the non impacted historical data, and impacted time series, obtained by taking, once selected a number of pre-event steps $T_{1}$ and post-event steps $T_{2}$ (with $T_{1} + T_{2} = T$), the windows that match this selection in the impacted historical time series.”
>
> As mentioned in the datasets section (5.1), it our dataset has 12 pre-event steps and 30 post-event steps, each step corresponding to a different month. Each time step in the real-world dataset corresponds to a univariate monthly sales volume for a product–country pair. Regarding the sample size, we included 2310 time series with positive event and 3000 time series with no event. We included this information in the updated version.
>
> As for the comment: “It also seems like a strong assumption that the event always occurs at the same time point or that the time series has the same length always”, we refer to same paragraph in 4.1 pasted above. We are not really assuming that the event will occur always at the same time step. Instead, we assume that we will always have some number of observations previous to the event, in such a way that we can always take a section of the time series that fits the setting.

---

> > ### Author Response · Authors · 2025-12-15
> >
> > Finally, with respect to the question of what question are we trying to answer by applying our method to the real world dataset if we do not have access to the ground truth, we aim to demonstrate the applicability of our model to real world datasets, where counterfactual ground truths are not available, showing also a complete protocol for evaluating desirable properties of counterfactual estimations in those cases.
> >
> > - **Why disentanglement is important/Theoretical claims**
> >
> >
> >     The disentanglement requirement in our model comes from the objective of making the latent variable behave as similarly as possible to the exogenous noise in structural causal models. As explained in Section 3.1, SCMs assume that all exogenous noises are mutually independent. Since each observed variable is a deterministic function of its parents and its own exogenous noise, this implies in particular that the exogenous noise of a variable is independent of its parents. In our implementation, the latent representation plays the role of such an exogenous noise; therefore, we require it to be approximately disentangled from the conditioners, i.e., independent of the variables that act as its parents in the graph. This is also emphasized in the paragraph *The importance of achieving a disentangled latent representation*. There are several works in image attribute editing ([1, 2, 3, 4]) that discuss the importance of this disentanglement, although not all of them use explicitly the Pearl framework. Other important works like DeepSCM [5] do not address this problem. However, in the conclusion, the authors of this work say: “further work should study more closely the training dynamics of deep mechanisms in SCMs: while not observed in our experiments, neural networks may not learn to cleanly disentangle the roles of its inputs on the output as expected. This could call for custom counterfactual regularisation, similar to losses used in image-to-image translation and explainability”. In our view, disentanglement is fundamental to achieve proper counterfactual estimations. Importantly, this is not an identification guarantee: as mentioned in the last paragraph of 4, there cannot be identifiability guarantees in multidimensional data.
> >
> >     The theoretical contribution of the paper is not a formal identifiability theorem, but an information-theoretic argument that, under a fixed reconstruction level, penalizing the entropy of the latent encourages Z to discard information already encoded in the conditioners C, and therefore to approximate the independent exogenous noise U_Y.
> >
> >     [1] Lample, G., Zeghidour, N., Usunier, N., Bordes, A., Denoyer, L., & Ranzato, M. A. (2017). Fader networks: Manipulating images by sliding attributes. *Advances in neural information processing systems*, *30*.
> >
> >     [2] He, Z., Zuo, W., Kan, M., Shan, S., & Chen, X. (2019). Attgan: Facial attribute editing by only changing what you want. *IEEE transactions on image processing*, *28*(11), 5464-5478.
> >
> >     [3] Liu, A. H., Liu, Y. C., Yeh, Y. Y., & Wang, Y. C. F. (2018). A unified feature disentangler for multi-domain image translation and manipulation. *Advances in neural information processing systems*, *31*.
> >
> >     [4] Cetin, I., Stephens, M., Camara, O., & Ballester, M. A. G. (2023). Attri-VAE: Attribute-based interpretable representations of medical images with variational autoencoders. *Computerized Medical Imaging and Graphics*, *104*, 102158.
> >
> >     [5] Pawlowski, N., Coelho de Castro, D., & Glocker, B. (2020). Deep structural causal models for tractable counterfactual inference. *Advances in neural information processing systems*, *33*, 857-869.
> >
> > - **Ablation without entropy penalty**
> >
> >     A conditioned regular Autoencoder without any mechanism to disentangle the latent representation from the conditioners produces extremely bad results, as the conditioner is mostly ignored. We have made this ablation for the non-confounded semi-synthetic dataset with positive factual event (semi-synth. 1): we got a MAE of 0.256 $\pm$0.009 and a MBE of -0.250$\pm$ 0.008. This is much worse than the other baselines (for reference, the best performing baseline in MAE is CEPAE, with .056 ± .004, and the worst was CVAE, with .083 ± .004). Furthermore, we see huge bias, which is due to the fact that, when using an autoencoder with no regularization, and use it to produce counterfactuals by changing the conditional value, it still performs mostly a reconstruction of the factual variable instead of a proper counterfactual. Thus, the entropy penalty is fundamental in CEPAE. We have added this result to the paper (5.4).

---

> ### Author Response · Authors · 2025-12-15
>
> **Under what conditions is minimizing this upper bound effective?**
>
> Regarding the entropy penalty, we do not claim a formal identification theorem, but there is a clear information-theoretic rationale. Our loss trades off (i) a reconstruction term, which encourages $(Z,C)$ to be informative about $X$, and (ii) a penalty on the entropy (capacity) of $Z$. Intuitively, if some aspect of $X$ can already be predicted from $C$, then encoding that same information in $Z$ is “wasteful” under the entropy penalty. The optimal compromise is therefore to let $C$ carry the information about $X$ that is explainable by $C$, and to use $Z$ primarily for the residual, exogenous variation. In other words, minimizing the entropy of $Z$ under a fixed reconstruction level encourages exactly the kind of disentanglement from the conditioner that is required by our SCM interpretation.
>
> It is true, as the reviewer says, that we are optimizing an upper bound on the entropy of $Z$, not the entropy itself. The mechanism works under typical conditions where (i) the bound is reasonably tight, and (ii) the model has enough capacity to exploit $C$ in the decoder. Failure modes could include, for example, choosing $\lambda$ (the weight of the entropy term) so large that $Z$ collapses and reconstructions degrade. We will add a short discussion of these conditions and limitations, as well as an ablation where we remove the entropy penalty (i.e., train a conditional autoencoder with only reconstruction loss) to empirically show that, in our experiments, the entropy-based disentanglement is indeed important for accurate counterfactual estimation.
>
> Finally, we would like to stress that optimizing tractable bounds on information-theoretic quantities is a completely standard practice in modern deep generative modelling and representation learning. VAEs, for example, do not maximize the exact log-likelihood but an evidence lower bound (ELBO); yet the ELBO has proved to be a reliable surrogate objective in practice. Similarly, many information-bottleneck and contrastive learning methods rely on computable upper or lower bounds on mutual information or entropy, and the guarantees are stated at the level of the bound rather than the intractable true quantity. Our entropy regularizer should be understood in exactly this spirit. The penalty we use is a monotone upper bound on the entropy $S(Z)$: shrinking the per-dimension scale of $Z$ always shrinks the bound and, under mild regularity assumptions on the latent distribution, also shrinks the true entropy. Thus, even though we do not optimize $S(Z)$ exactly, the optimization signal is aligned with the desired direction (lower latent capacity). As our ablation shows, this surrogate is sufficient in practice to obtain the degree of disentanglement needed for accurate counterfactual estimation.
>
> **Why only autoencoders are considered in this work?**
>
> Our focus on autoencoders is mainly conceptual: they give a very direct and transparent implementation of the abduction–action–prediction pipeline we use, where the encoder naturally plays the abduction step (inferring a latent $Z$ from $(H,E,Y)$, which we interpret as a learned exogenous noise) and the decoder implements the structural equation $Y=f_Y(H,E,Z)$, so that the action–prediction step reduces to reusing the same $Z$ while changing only $E$ to obtain a counterfactual trajectory. For this, we need a model with an explicit encoder from data to latent and an explicit conditional decoder from $(H,E,Z)$ to $Y$; autoencoders (AE/CVAE-type models) provide exactly this structure with minimal extra machinery, which keeps the SCM–model correspondence simple and allows a clean comparison between CAAE, CVAE, and our entropy-regularized CEPAE. While normalizing flows do offer a straightforward invertible mapping between data and latent space in principle, designing flows that are both efficient and expressive for high-dimensional, structured time-series blocks is considerably more involved, often requiring architectural constraints for tractable Jacobians and leading to a different set of trade-offs; similarly, GANs or diffusion models would add substantial architectural and optimization complexity that could obscure the specific effect of our entropy-based regularization, and they do not provide a straightforward way to invert data and perform the abduction step. For these reasons, in this work we restrict ourselves to autoencoder-based models as a widely used and stable instantiation of conditional latent-variable models, while noting that extending our framework to normalizing flows or other generative families is an interesting direction for future work.

---

> ### Author Response · Authors · 2025-12-15
>
> **Discussion on the main results table and metrics**
>
> We consider MAE to be the most important metric because it directly compares the counterfactual estimations with the ground truth in terms of the absolute error. MBE measures the mean value of the errors (not absolute errors): $\mathrm{MBE} = \frac{1}{n}\sum_{i=1}^{n} (\hat{y}_n - y_n)$, and it’s objective is to show if there are biases in the counterfactual estimation (i.e., if estimations are systematically bellow or above the ground truth). This is interesting in our context because, when a model based on abduction-action-prediction not use the conditioner, and the latent representation brings information shared with the conditioner, the estimated counterfactuals tend to be closer than they should to the factual observations. An extreme example of that is the Autoencoder without entropy penalty, which has a MBE so distant from 0.
>
> Whereas in MAE, CEPAE outperforms the baselines in all datasets, in MBE the LSTM and AB-LSTM slightly outperform it. This is expected because LSTM and AB-LSTM are standard forecasting models rather than abduction–action–prediction models: they are trained to predict the post-event series directly from $(H,E)$ and are therefore not biased towards the factual trajectory in the same way. However, among the abduction–action–prediction models (CVAE, CAAE, CEPAE), CEPAE achieves the best or comparable MBE on all datasets except Synthetic 0, where the difference with CAAE is very small. This supports our claim that the entropy penalty helps to reduce the bias introduced when the latent still encodes information that should be carried by the conditioner.
>
> The rest of the metrics (Added Variations and the axiomatic metrics) complement MAE/MBE by probing structural properties of the counterfactual mapping. Together, the are a good way to evaluate counterfactuals for real world datasets, where we do not have ground truth, and we use them on synthetic and semi-synthetic datasets for completeness. The Added Variations metrics measure how the estimated counterfactual responds to local perturbations in the factual post-event series. For each series, we add controlled positive and negative perturbations on short windows of the factual post-event trajectory, then recompute the counterfactual and compare it to the unperturbed counterfactual. We report (i) Total difference, which compares the two counterfactuals over all time steps, and (ii) Altered steps difference, which compares them only on the perturbed steps. Both are normalised so that the ideal value is 1. Intuitively, a good model should propagate these local variations to the counterfactual estimate—because they are due to idiosyncratic noise, not to the event—without unnecessarily amplifying or suppressing them. In Table 1, CEPAE is consistently closer to the ideal value 1 than the other autoencoder-based models across datasets, which indicates that it better preserves such local variations in a way that is coherent with the underlying SCM. This is particularly visible in the confounded synthetic setting, where models that do not disentangle the latent from the conditioner either under-react (values substantially below 1) or over-react (values above 1) to the injected perturbations.

---

> > ### Author Response · Authors · 2025-12-15
> >
> > The axiomatic metrics—Reconstruction, Reversibility and Effectiveness—follow Monteiro et al. (2023) and evaluate counterfactual soundness without requiring ground-truth counterfactuals. Reconstruction (lower is better) measures whether “null interventions” leave the observation unchanged; in our setting, this means applying the counterfactual mechanism but keeping parents $(H,E)$ factual. Reversibility (lower is better) checks whether applying the mapping forward (from factual to counterfactual parents) and then backward (swapping parents again) approximately recovers the original observation. Effectiveness (higher is better) checks that intervening a parent (here, the event) actually changes the estimated counterfactual in the direction predicted by a pseudo-oracle for that parent. On the synthetic and semi-synthetic datasets, CEPAE generally has the lowest Reconstruction and Reversibility and competitive or slightly lower Effectiveness compared to CVAE, which indicates that CEPAE’s counterfactual mapping is closer to respecting the Pearlian axioms while still reacting strongly to the event. On the real-world dataset, where no counterfactual ground truth is available, MAE/MBE cannot be computed (hence the “–” entries in the top block of Table 1), and our evaluation necessarily relies on Added Variations and axiomatic metrics; there CEPAE again matches or improves upon the other models, suggesting that the benefits of the entropy-regularized latent also transfer to this more realistic setting. Regarding the missing CAAE values in Table 1: as discussed in Section 4.2.1, CAAE trains reliably in the unconfounded datasets but we were unable to find a hyperparameter configuration that yields stable and meaningful results in the confounded scenario. In those cases the adversarial training either collapses or produces degenerate counterfactuals, so we chose not to report those numbers and instead mark the corresponding entries as “–” to avoid over-interpreting an unstable model.
> >
> > We did not include MSE because it measures basically the same thing as MAE: how counterfactual estimations compare to ground truths. While we give many metrics, all of them measure a particular and different aspect of the counterfactual estimations. We found that MSE induces essentially the same ranking as MAE across models and datasets; we therefore kept MAE in the main table for readability. Moreover, MAE is the base distance used in the axiomatic metrics (Reconstruction, Reversibility), so using MAE keeps all metrics more consistent.
> >
> > We have added a definition of MBE in the metrics paragraph, and have extended the results section to further comment the trends of table 1.
> >
> > We hope that our additions and clarifications address your concerns and make the contributions and limitations of our work clearer, and we thank you again for your constructive feedback.

---

> > > ### Comment · Reviewer_WBjb · 2025-12-18
> > > **Response to authors**
> > >
> > > I thank the authors for their detailed responses and for providing additional experimental results for the ablation study I requested. I appreciate the more in-depth discussion of the additional metrics in the experiments, although the current presentation of the results is somewhat dense and hard to read. I've two remaining things which I would like to mention.
> > >
> > > First, I am still not fully convinced by the reply regarding the time-series aspect of the problem. In particular, I find the following statement in your response unclear:  “We are not really assuming that the event will occur always at the same time step. Instead, we assume that we will always have some number of observations prior to the event, such that we can always extract a segment of the time series that fits the setting.” Under this assumption, is there also an implicit requirement that the time interval between consecutive data points is always equal?
> > >
> > > Second, after rereading the paper and looking at the other reviews, I agree with the concern that the presentation needs improvement. Even in the revised version, I would say that these issues persist. The overall exposition would benefit from a more streamlined and focused structure, especially because I find the core ideas to be diluted across Section 4 and the current experiment section hard to read. At this point, this is my main hesitation regarding acceptance. I would find it helpful to know whether the other reviewers see the revised presentation as an improvement, since I may be more lenient if the other reviewers do not share my concern.

---

> > > > ### Author Response · Authors · 2025-12-18
> > > >
> > > > Thank you for the additional feedback and for taking the time to reread the revised manuscript. We address your two remaining points below.
> > > >
> > > > **(1) Time-series sampling assumption (equal interval).**
> > > >
> > > > To your question: “Under this assumption, is there also an implicit requirement that the time interval between consecutive data points is always equal?” Yes—as currently formulated, our setting assumes regularly sampled time series, i.e., a constant time interval between consecutive observations (within each extracted window). This is the regime our paper targets, and it matches all datasets used in the paper: the semi-synthetic dataset consists of daily sales and the real-world dataset of monthly sales, both on a regular grid. We have now made this assumption explicit in Sec. 4.1.
> > > >
> > > > **(2) Presentation: making Sec. 4 and the Results section less dense / less diluted.**
> > > >
> > > > We agree that the exposition can be easier to follow. In the previous revision we primarily improved the prose (tone, concision, and precision); in this revision we additionally improved the structure and signposting so that the core ideas and empirical conclusions are easier to extract.
> > > >
> > > > **Results section (Sec. 5.4).** We reworked the presentation to clearly separate metric families and evaluation goals:
> > > >
> > > > - Added **Key takeaways** bullet points at the start of the Results section to summarize the main conclusions before the detailed discussion.
> > > > - Reformatted the main results table to clearly separate three metric groups: Ground-truth metrics, Added Variations, and Axiomatic metrics.
> > > > - Rewrote the discussion of results into short, clearly titled paragraphs aligned with those metric groups and the additional synthetic-control comparison:
> > > >     - *Ground-truth metrics*
> > > >     - *Ablation and computational cost*
> > > >     - *Proxy metrics: Added Variations and axiomatic metrics*
> > > >     - *Comparison with Synthetic Control under controlled donor quality*
> > > >
> > > > **Method section (Sec. 4).** We added several navigation aids to reduce cognitive load and make the pipeline explicit:
> > > >
> > > > - Added a short **roadmap** paragraph at the beginning of Sec. 4.2 that states the structure of the subsection and points to the central definitions.
> > > > - Added **Box 1 (“Method in three steps”)** summarizing the abduction–action–prediction procedure implemented by the encoder–decoder models, so the core counterfactual inference pipeline is visible immediately without requiring reconstruction from surrounding text.
> > > > - Kept Table 1 (model comparison) as the anchor for the differences between CVAE, CAAE, and CEPAE.
> > > >
> > > > Overall, we believe Sec. 4 is now clearly partitioned by function: Sec. 4.1 defines the setting and assumptions; Sec. 4.2 introduces the encoder–decoder abduction–action–prediction template and then presents CVAE/CAAE/CEPAE as instantiations, with Box 1 summarizing inference and the roadmap guiding the reader. Within Sec. 4.2, the three models are described in separate, clearly labeled parts, each as a concrete instantiation of the same template, and Table 1 concisely summarizes their differences.
> > > >
> > > > We are happy to make further presentation edits if you can point to specific parts that still feel hard to follow. We believe the technical contribution and empirical evidence are solid, and we would like the final decision to reflect the substance rather than avoidable clarity issues.

---

> > > > > ### Comment · Reviewer_WBjb · 2025-12-21
> > > > > **Final reply to authors**
> > > > >
> > > > > Once again I want to thank the authors for their reply and the new revision.
> > > > >
> > > > > I feel like all of my content-level concerns have been addressed. I've the same feeling as reviewer CdHW that the manuscript ultimately would have benefited from being shorter than it is now, as it would improve readability. But this shouldn't be a reason for rejection in my opinion, especially because the earlier issues with clarity seem to be fixed. Thus I'm in favour of recommending accept for this manuscript.

---

### Review · Reviewer_CdHW · 2025-12-08

**Summary Of Contributions:**

The authors propose CEPAE, a novel framework that uses entropy regularization in latent space towards improving counterfactual inference for time series.

They compare their approach to two established models (CVAE and CAAE) across a range of synthetic, semi-synthetic and a real-world dataset(s). Given the considered metrics, CEPAE exhibits favourable performance w.r.t. competing approaches in the majority of tasks.

While the empirical evidence suggests an advantage of the extensions that the authors propose, these are often presented in a way that makes it hard to discern *why* these extensions offer a benefit and how they relate to prior work.

**Additional Comments:**

The reference for the VAE is incorrect. It was published in 2013. Please use the non-arxiv version.

**Audience:**

Yes

**Audience Explanation:**

There is a large body of work on counterfactual inference, as well as a large community in time series causal inference so there should be interest from both communities in this work.

**Broader Impact Concerns:**

-

**Claims And Evidence:**

No

**Claims Explanation:**

My main concern with this manuscript lies in the presentation.

As a general comment, I feel that the contents could be compressed which would also greatly help the clarity of the message the authors wish to convey. Many passages are written almost colloquially, leading to a lack of precision and lack of clear thread. In other sections the number of commas used disrupts the flow of reading. The citation style is inconsistent, with inline citations often being used where standard citations should be used. While these are style issues that can be immediately improved, there are many instances when the authors argues in hand wavy fashion, leading to doubts about the authors claims.

Specifically, when discussing related works, the authors allude to a specific connection or result with explicitly stating what they mean. This happens throughout, but to single out an example in Section 4.2. they state "[f]or the problem of counterfactual estimation in high dimensional settings, autoencoder based approaches are the most extended and effective." without backing up this claim with any reference. This continues in the next paragraph, claiming that $\mathbf{Z}$ is equivalent to $U_y$ and immediately stating that the noise is not identifiable. This apparent contradiction is never explicitly resolved. The next paragraph about the importance of disentangled representations is also hand wavy and not backed up by enough precise theoretical statements.

To repeat: the empirical results suggest that the authors' contributions are useful and the arguments the authors do provide don't seem completely off, but the presentation is not up to par for a scientific paper.

**Requested Changes:**

The paper should be rewritten to be more concise and accurate in the stated claims. If claims can be made in a theoretically sound manner, they should be done this way and if not, they should not be alluded to in hand wavy, speculative fashion, but rather not made. The empirical evidence suggests that the authors are on to something, but the presentation is lacking for publication in this state.

---

> ### Author Response · Authors · 2025-12-15
>
> We thank the reviewer for the careful reading of our manuscript and for acknowledging the technical interest and empirical usefulness of our contributions. We apologize for the imprecisions and the colloquial tone in the original submission. In the revised version we have substantially rewritten several parts of the text to make the presentation more concise, precise, and theoretically well-grounded. Below we summarize the main changes addressing your concerns.
>
> **1. Clarifying the role of $Z$ and the SCM analogy (Sec. 4.2)**
>
> You pointed out that the relationship between the latent variable $\boldsymbol{Z}$ and the exogenous noise $U_y$ in the SCM was confusing and hand-wavy, especially in the paragraph following “For the problem of counterfactual estimation in high dimensional settings…”. We have now rewritten this paragraph to make the abduction–action–prediction analogy explicit and to clearly state that $\boldsymbol{Z}$ is only a proxy for the exogenous noise and not an identifiable quantity. The problematic paragraph has been replaced by:
>
> “There is a direct analogy between this encoder--decoder setting and the abduction--action--prediction procedure. In SCM terms, abduction amounts to inferring a distribution over the exogenous noise $U_{y}$ given $(\boldsymbol{h}, e_{f}, \boldsymbol{y_f})$, i.e., $P{\mathcal{M}}(U_{y} \mid \boldsymbol{h}, e_{f}, \boldsymbol{y_f})$. Our encoder plays the role of learning a latent variable $\boldsymbol{Z}$ that summarizes this noise: it aims to represent the part of the variation in $\boldsymbol{y}$ that is not explained by $(\boldsymbol{h}, e{f})$. Importantly, following impossibility results on the identifiability of generative mechanisms with multidimensional exogenous variables \citep{nasr2023impossibility,monteiro2023measuring}, the true exogenous noise $U_{y}$ is not identifiable in general. Therefore, we do not claim that $\boldsymbol{Z}$ equals $U_{y}$; instead, we only require that $\boldsymbol{Z}$ is a useful proxy that behaves like such a noise term for the purpose of counterfactual prediction. Then, the decoder (that can be identified with the structural assignment $f_{y}$) inferring $\hat{\boldsymbol{y_{cf}}}$ corresponds to the action and prediction steps, where we first set the modified SCM $\widetilde{\mathcal{M}}= \mathcal{M}{\boldsymbol{h},e_{f}, \boldsymbol{y}{f}; , do(E=e_{cf})}$ and then obtain $\hat{\boldsymbol{y_{cf}}}$ as a sample from $\widetilde{\mathcal{M}}$.”
>
> This directly addresses the apparent contradiction you pointed out and ties the method more tightly to the SCM semantics.
>
> **2. Softening and properly supporting claims about autoencoders (Sec. 4.2)**
>
> You were concerned about strong and insufficiently supported claims such as “autoencoder based approaches are the most extended and effective”.
>
> We have removed this statement and replaced it with a more cautious and referenced formulation:
>
> “For counterfactual estimation in high-dimensional settings, autoencoder-based architectures are a natural choice as they provide flexible latent representations (Pawlowski et al., 2020; Sanchez-Martin et al., 2021;Lample et al., 2017). In this work, we adapt two such architectures to our time-series setting and propose a new one.”
>
> **3. Clarifying the disentanglement argument and the role of the entropy penalty (Sec. 4.2)**
>
> You described the paragraph on the importance of disentangled representations as “hand wavy and not backed up by enough precise theoretical statements”. We have rewritten this part to clearly state the modelling intuition in terms of the SCM and to connect it with the information-theoretic arguments in Sec. 4.2.3.
>
> For example, we now explain that if the conditioners are informative about $X$, then encoding the same information in $\boldsymbol{Z}$ is redundant and penalized by the entropy term, thereby encouraging $\boldsymbol{Z}$ to capture only variability analogous to $U_y$. This replaces earlier, more informal phrasing and is explicitly grounded in the mutual-information analysis presented later in the section.

---

> > ### Author Response · Authors · 2025-12-15
> >
> > **4. Shortening and simplifying long sentences**
> >
> > You noted that many passages were written in a colloquial style with long sentences and many commas, which hindered clarity. We carefully went through the main text and split several long sentences into shorter ones, removed colloquial turns of phrase, and simplified wording.
> >
> > For instance, we changed the following sentence in the problem-setting section:
> >
> > “$\boldsymbol{H}$ and $E$ will be always intervened ($\boldsymbol{H}$ to its same factual value $\boldsymbol{h}$ and $E$ to the counterfactual value $e_{cf}$). Therefore, only for the variable $\boldsymbol{Y}$ it will be necessary to estimate a structural assignment $f_{y}$ and to abduct its exogenous noise $U_{y}$, responsible for its variation once given $E$ and $\boldsymbol{H}$. Thus, it is not necessary to treat $E$ and $\boldsymbol{H}$ as random variables generated by an exogenous noise susceptible to be abducted (and, in the confounded case, generated also by their parents).” to the more concise and direct: ”In our counterfactual setting, $E$ and $H$ are always intervened on ($H$ to its factual value $h$ and $E$ to the counterfactual value $e_{\mathrm{cf}}$). Therefore, it is not necessary to treat $E$ and $H$ as random variables generated by exogenous noise terms that must be abducted.”
> >
> > Similarly, in the datasets/metrics section we replaced a very long sentence:
> > ”Two alternatives exist to overcome this issue: the use of metrics that do not directly measure similarity among counterfactual estimations and ground truths but desirable properties of counterfactuals, which we address in 5.2, and the use of synthetic or semi-synthetic datasets, where we control the data generating process, or at least a part of it, and therefore have access to the counterfactual ground truths, which allow to apply traditional metrics.”
> > by: ”Two alternatives are commonly used to address this issue. First, one can employ metrics that do not directly measure similarity between counterfactual estimates and ground truths but instead evaluate desirable properties of counterfactuals; we discuss these in the metrics section. Second, one can use synthetic or semi-synthetic datasets, where the data-generating process is (at least partially) under control and ground-truth counterfactuals are available, allowing traditional error metrics to be computed.”
> >
> > We made similar edits throughout the main text (introduction, related work, model description, and experiments) to improve readability.
> >
> > **5. Citation style and minor presentation issues**
> > You also pointed out inconsistencies in citation style and an incorrect reference for VAEs. In the revised version:
> >
> > - We standardized the use of author–year citations (e.g., using “\citet” when the authors are the subject and “\citep” otherwise).
> > - We corrected the VAE reference to the original, non-arXiv publication by Kingma and Welling.
> > - We removed informal expressions and subjective adjectives (e.g., “very weak methodology”, “seizes information”) and replaced them with more neutral and precise wording.
> >
> > We hope these changes address your concerns about the clarity and precision of the presentation. We believe the revised manuscript now better communicates the underlying ideas and their limitations, while preserving the empirical strengths you highlighted.
> >
> > We thank the reviewer again for their constructive comments and remain open to further suggestions for improvement.

---

> > > ### Comment · Reviewer_CdHW · 2025-12-19
> > > **Response to authors**
> > >
> > > I thank the authors for their detailed response to my comments as well as their provided revisions.
> > >
> > > The presentation has improved in my opinion, as well as the clarity of certain sections. I also appreciate the ablations provided for the comments of other reviewers, as these help clarify one of my original doubts as to *why* the proposed approach is beneficial.
> > >
> > > While my main concerns regarding presentation and hand-waviness of claims have been addressed, I still feel like the contents of this work do not need to be spread over 13+ pages. Given the verbosity of responses, but also the speed in which substantially revised versions of the manuscript are presented, I cannot help but have the impression that the authors are heavily relying on an LLM for these edits. While I do not object such tools in general, I do invite the authors to take their time to think about which sections may be compressed or shortened. Now the paper seems almost bloated with introductory paragraphs and exposition sentences. It is ok to simply show the reader concisely what the message is, instead of first paraphrasing it. I believe the paper would benefit from being constrained to fewer pages.
> > >
> > > I will wait on the responses of the other reviewers to give my final recommendation, but am generally leaning towards acceptance.

---

> > > > ### Author Response · Authors · 2025-12-19
> > > >
> > > > Thank you for the follow-up and for your improved assessment after our revisions. We agree the manuscript can be further compressed, and that doing so would improve readability. We also agree this is best done with a careful pass rather than rushed edits in the final days of the discussion period. If accepted, we will streamline the final version by removing redundant exposition, tightening Section 4, and moving secondary background material to the appendix. Thank you again for the constructive suggestion.

---

> > > > > ### Comment · Reviewer_CdHW · 2025-12-22
> > > > > **Final response**
> > > > >
> > > > > I thank the authors for their reply. I agree with the other reviewers and now also suggest acceptance.

---

### Comment · Reviewer_ah7J · 2025-11-03
**Interesting Work but Need More Improvements**

This manuscript presents a new framework for counterfactual inference in time series data, introducing the Conditional Entropy-Penalized Autoencoder (CEPAE). The study tests CEPAE on synthetic, semi-synthetic, and real-world datasets, comparing it against established baselines including LSTM forecasts, CVAE, and CAAE.

This paper makes a meaningful contribution to the intersection of causal inference, time series modeling, and representation learning. However, there are several ways that this paper needs to improve:

While the authors acknowledge the challenges of comparing against synthetic control or causal impact methods, this omission weakens the empirical claim of superiority. Even limited comparisons (e.g., using standard Bayesian Structural Time Series) could provide stronger grounding for CEPAE’s advantages in practice.

The manuscript acknowledges the impossibility of full identifiability in multidimensional counterfactuals (as per Nasr-Esfahany & Kiciman, 2023). Nonetheless, it would benefit from a more explicit discussion on the interpretability of how the learned latent variables relate to domain factors and whether CEPAE’s disentanglement produces semantically meaningful components.

While the paper notes that CEPAE avoids adversarial instability, there is little quantitative discussion of its computational cost, convergence behavior, or hyperparameter sensitivity. These are important factors for reproducibility and industrial scalability.

[Optional] The real-world evaluation lacks ground-truth counterfactuals, making interpretive confidence somewhat limited. Although the authors employ proxy metrics, a qualitative example or case study illustrating CEPAE’s behavior in an industrial dataset would substantively strengthen the paper’s empirical narrative.

The exposition is dense and assumes substantial background in both causal inference and information theory. Several theoretical sections could be streamlined or visually summarized to improve readability. The core conceptual differences among CVAE, CAAE, and CEPAE could also be more distinctly contrasted.

---

### Decision · Action_Editor_aUez · 2026-01-20

**Recommendation:** Accept as is

**Additional Comments:**

The paper has gone through a substantial revision to make the reviewers' requested changes, primarily related to the clarity of the paper's claims and overall presentation. After the revision, all reviewers leaned toward accepting the paper, as do I.

**Audience:**

Yes

**Audience Explanation:**

Counterfactual inference is a problem of continued interest in the TMLR community.

**Claims And Evidence:**

Yes

**Claims Explanation:**

Reviewers were unanimous in their view that this paper presents clear evidence for its claims.